# Leveraging explainable artificial intelligence for early prediction of bloodstream infections using historical electronic health records

**Rajeev Bopche** [1]*, **Lise Tuset Gustad** [2,3], **Jan Egil Afset** [4], **Birgitta Ehrnström** [5,6,7], **Jan Kristian Damås** [5,6], **Øystein Nytrø** [1,8]

**1** Department of Computer Science, Norwegian University of Science and Technology, Trondheim, Norway, **2** Faculty of Nursing and Health Sciences, Nord University, Levanger, Norway, **3** Department of Medicine and Rehabilitation, Levanger Hospital, Nord-Trøndelag Hospital Trust, Levanger, Norway, **4** Department of Medical Microbiology, St. Olavs Hospital, Trondheim University Hospital, Trondheim, Norway, **5** Department of Clinical and Molecular Medicine, Norwegian University of Science and Technology, Trondheim, Norway, **6** Department of Infectious Diseases, Clinic of Medicine, St Olavs Hospital, Trondheim, Norway, **7** Clinic of Anaesthesia and Intensive Care, St Olavs Hospital, Trondheim University Hospital, Trondheim, Norway, **8** Department of Computer Science, The Arctic University of Norway, Tromsø. Norway

\* rajeev.bopche@ntnu.no

**Data Availability Statement:** We have made the code and data used in this study available on a public repository. This includes all scripts for data preprocessing, data enhancement, model training,

## Abstract

Bloodstream infections (BSIs) are a severe public health threat due to their rapid progression into critical conditions like sepsis. This study presents a novel eXplainable Artificial Intelligence (XAI) framework to predict BSIs using historical electronic health records (EHRs). Leveraging a dataset from St. Olavs Hospital in Trondheim, Norway, encompassing 35,591 patients, the framework integrates demographic, laboratory, and comprehensive medical history data to classify patients into high-risk and low-risk BSI groups. By avoiding reliance on real-time clinical data, our model allows for enhanced scalability across various healthcare settings, including resource-limited environments. The XAI framework significantly outperformed traditional models, particularly with tree-based algorithms, demonstrating superior specificity and sensitivity in BSI prediction. This approach promises to optimize resource allocation and potentially reduce healthcare costs while providing interpretability for clinical decision-making, making it a valuable tool in hospital systems for early intervention and improved patient outcomes.

## Author summary

In this research, we have developed a new tool that uses artificial intelligence to better predict bloodstream infections, which can lead to severe conditions like sepsis if not quickly identified and treated. It is the first-of-its-kind framework that analyzes past health records and helps identify patients at high risk of infection more accurately than existing tools. Unlike existing tools, our framework can be implemented at any stage of the patient trajectory and is the only framework to achieve good accuracy without the use of intimate patient features such as vital signs and real-time data, which may limit clinical applicability. This ability could enable doctors to prioritize care more pre-emptively and effectively,

evaluation, interpretations, and the case study. This study's code, outputs/results, and data statistics are available at https://github.com/EngineerRajeev/Advancing-Bloodstream-Infection-Prediction-Using-Historical-Electronic-Health-Records-.git to foster reproducibility. The raw de-identified electronic health records used are managed by St. Olavs Hospital in Trondheim, Norway. Due to the sensitive nature of the raw data, it cannot be made publicly available. To request data access for replication, please send an email to the Head of the department at forskningsavdelingen@stolav.no. The guidelines are provided by the Ethics Oversight Board and Data Access Committee St. Olavs Hospital in Trondheim, Norway.

**Funding:** The Computational Sepsis Mining and Modelling project provided financial support for this study through the Norwegian University of Science and Technology Health Strategic Area. The funders had no role in study design, data collection and analysis, decision to publish, or preparation of the manuscript.

**Competing interests:** The authors have declared that no competing interests exist.

potentially saving lives and reducing unnecessary medical tests. Our approach is designed to be easily understood and used by medical professionals and those with little technical expertise, making it a valuable addition to hospital systems.

# 1. Introduction

Bloodstream infections (BSIs) are a significant public health challenge, often leading to severe clinical conditions such as sepsis and septic shock, particularly when unrecognized or untreated. The rapid progression of these infections, coupled with their association with high morbidity, mortality, and healthcare costs, renders BSIs a critical challenge in clinical care [1,2]. The accuracy of available clinical decision tools for BSI and sepsis needs improvement. Most of them are based on changes in vital signs and abnormal blood test results [3,4]. Improved prediction can lead to more efficient allocation of resources and reduced health costs. A refined initial stratification would aid in allocating resources to patients with a high risk of a BSI and reduce needless testing of patients with low risk. Blood culture (BC) may yield relevant bacteria causing disease and the growth of contaminating microbes. Ambiguous culture results may be difficult to interpret and lead to clinical uncertainty, often resulting in more extended hospital stays and unnecessary administration of antibiotics [5,6]. Therefore, reduced collection of BC in patients with a low risk of BSI can lead to a higher positive predictive value of blood cultures and reduced use of antibiotics. Prompt and early identification of high-risk versus low-risk patients is thus imperative for adequate and effective initial handling of suspected BSI, optimized allocation of healthcare resources, and reduced associated costs.

The advent of artificial intelligence (AI) allows innovative methodologies for BSI diagnostics, showcasing the potential to enhance or even surpass human expertise in diagnostic endeavors [7]. Despite its demonstrated efficacy, the integration of AI into clinical workflows remains limited [8,9]. Facilitating this integration may involve leveraging AI models to augment guidelines-based clinical decision support systems (CDSS) rather than striving to develop fully autonomous AI-based CDSS [10,11]. Furthermore, adopting a stance grounded in predictive, preventive, and personalized medicine (PPPM) principles could refine the utilization of AI, emphasizing the analysis of historical rather than real-time data variables [12,13]. The primary aim of this study was to develop and validate an eXplainable Artificial Intelligence (XAI) framework that utilizes historical electronic health records (EHRs) to enhance the prediction of BSIs, thus facilitating precise intervention. By integrating a historical dataset with advanced machine learning (ML) techniques, we sought to overcome the limitations of current diagnostic methods. We presents the XAI-based BSI prediction (XBSI) framework for early prediction of BSI. Our findings confirm that the XBSI framework meets and exceeds traditional models demonstrating enhanced predictive accuracy and interpretability.

## 1.1. Literature review

BSI is a critical precursor to sepsis, a severe and potentially life-threatening condition. Early detection and management of BSI can significantly mitigate the risk of progressing to sepsis. In this context, leveraging AI models to predict and manage BSI presents a promising avenue. Our literature review sought to capture the breadth and heterogeneity of recent advances in ML-based models for BSI prediction. We compiled and analyzed thirty studies published in the last five years, focusing on various healthcare settings and patient demographics [11,14–43]. These studies predominantly focused on inpatient settings, constituting 56% (n = 17) of the research, followed by emergency department (ED) settings at 23% (n = 7) and intensive

care unit (ICU) settings at 20% (n = 6). Within the inpatient group, the studies varied, with nine examining general populations [14–22], two targeting inpatients with central venous catheters (CVC) [23,24], and others focusing on specific patient categories such as hemodialysis (HD) patients [26], cancer patients [27], maternity patients [28], patients with low procalcitonin levels (PCT ≤2.0 ng/ml) [29], and human immunodeficiency viruses (HIV) infected patients [30]. Bacteremia was the primary condition under study in 24 articles, including varied focuses such as fungemia [14] and candidemia [27,42], while three studies aimed at predicting central line-associated bloodstream infections (CLABSIs) [32,33,40] and one on hospital-acquired BSI (HA-BSI) [22]. All articles reported high prediction performance (area under the receiver operating characteristic curve (AUROC) > 0.7) except for one article [20]. Most studies were single-centered, with three studies sourcing data from two hospitals [14,16,37] and two studies using data from multiple centers [34,42]. The key characteristics such as target condition, number of patients or samples, data source, prevalence, ML models, and top predictors for each study grouped by settings are given in Table 1.

Among studies with study design of inpatient settings, Bhavani et al. (2020) used EHRs from two academic tertiary medical centers between 2007 and 2018. Data types included demographic data, International Classification of Diseases (ICD) billing codes, clinician orders, BC results, vital signs, nursing assessments, and laboratory data [14]. The logistic regression (LR) and gradient boosting machine (GBM) models were developed. The GBM model demonstrated superior performance in predicting bacteremia and fungemia with AUROC scores of 0.78 for bacteremia and 0.88 for fungemia prediction. Lee et al. (2019) investigated the early detection of bacteremia using an artificial neural network (NN) model, specifically a multi-layer perceptron (MLP). The study utilized data from 13,402 patients at Gangnam Severance Hospital in South Korea, identifying 1,260 episodes of bacteremia from blood cultures [15]. Data included 20 clinical variables, such as vital signs and various laboratory data. The study highlighted the effectiveness of MLP models, showing remarkable sensitivity in identifying bacteremia episodes based on a well-curated set of clinical variables. Lee et al. (2022) expanded on earlier research, utilizing an extensive dataset from Gangnam Severance Hospital, consisting of a larger patient sample size and more recent patient admissions from 2013 to 2018 [16]. This study employed enhanced NN models alongside other ML techniques like Random Forest (RF) and Support Vector Machines (SVM) to predict bacteremia from clinical and laboratory data. The study by Mahmoud et al. (2021) developed a predictive model for bacteremia using data from 7,157 adult patients admitted to King Abdulaziz Medical City in Riyadh. This retrospective cohort study utilized EHRs from July 2017 to July 2019 to analyze 36,405 blood culture tests [20]. The dataset included demographics, clinical variables such as vital signs (e.g., temperature, heart rate, blood pressure), and laboratory data (e.g., white blood cell (WBC) count, platelet count, creatinine level, lactic acid level, C-reactive protein (CRP), and procalcitonin levels). Several ML models were employed to determine the best predictor of positive blood cultures, including Neural NN, RF, LR, Decision Trees (DT), Naive Bayes (NB), and SVM with a radial basis function (RBF) kernel. Garnica et al. (2021) utilized a Hospital Universitario de Fuenlabrada dataset in Madrid, comprising 4,357 patients with 117 features per patient [21]. The features included demographics, medical history, clinical analysis, comorbidities, and the results of blood cultures, differentiated into cases of bacteremia and no bacteremia. Three supervised ML classifiers were implemented: SVM, RF, and K-Nearest Neighbors (KNN). Each classifier was employed in two scenarios: using only the features available during blood extraction and a second scenario with additional features revealed during the blood culture process.

Among studies in the ED settings, Choi et al. (2022) developed ML models to predict bacteremia at Seoul National University Hospital [31]. The study used data from 24,768 adult

**Table 1. The key characteristics of the studies using ML for BSI prediction.**

| Setting | First author, year | Target condition | No. Patients | Data source | Prevalence | ML models | Key predictors |
|---------|-------------------|------------------|--------------|-------------|------------|-----------|----------------|
| Inpatients | Bhavani et al. (2020) [14] | Bacteremia and Fungemia | 76688 | EHRs, University of Chicago Hospital (2008–2018) and Loyola University Medical Center, USA (2007–2017) | Bacteremia 7.7%, Fungemia 0.7% | LR, GBM | Time from admission to BC, Temperature, Age, HR, Prior Bacteremia/Fungemia, WBC, BUN, Glucose, DBP, SBP, PPI, RR |
| | Lee et al. (2019) [15] | Bacteremia | 13402 | EHRs, Gangnam Severance Hospital, Seoul, Republic of Korea, (2008–2012) | 7.9% | MLP, SVM, RF | ALP, PLT, Temperature, SBP, WBC, ICU stay, CRP, CVC, Age, PT, Hospital days to BC, HR, Gender, Antibiotics, RR, Creatinine |
| | Lee et al. (2022) [16] | Bacteremia | 622771 samples | EHRs, Sinchon and Gangnam Severance Hospitals, Republic of Korea, (2007–2018) | 6.2% | MLP, RF, XGB | PLT, Monocyte, Neutrophil, Bilirubin, Albumin, and Hospital stay, BUN, ALP, RR, PR, DBP, TP, WBC, PT, Hb, CRP, Creatinine, ALT, AST, Sodium, Chloride, ESR |
| | Cheng et al. (2020) [17] | Bacteremia | 28043 | EHRs, Zhengzhou University Hospital, China, (2017–2018) | 10% | LR, NB, SVM, ADT, CNN, BiLSTM, ABiLSTM + DAE | Textual chief complaints, Admission records, and Laboratory biochemical indicators. |
| | McFadden et al. (2023) [18] | Bacteremia | 10965 samples | CBC/DC, CPD, Sir Charles Gairdner Hospital, Western Australia (2018–2020) | 7.58% | RF, XGB | CBC, DIFF, and CPD |
| | Lien et al. (2022) [19] | Bacteremia | 366586 samples | EHRs, CBC/DC, Linkou Chang Gung Memorial Hospital (CGMH) in Taiwan, (2014–2019) | 8.2% | RF, LR | CBC/DC, CRP, and PCT |
| | Mahmoud et al. (2021) [20] | Bacteremia | 7157 | EHRs, King Abdulaziz Medical City, Riyadh, Saudi Arabia (2017–2019) | 11.4% | NN, RF, LR, DT, NB, SVM | Age, Antibiotics use, Surgery within 14 days, CVC, length of hospitalization before BC, RR, SBP, Temperature, DBP, HR, WBC, Sodium, PLT, Albumin, Creatinine, Lactic acid level. |
| | Garnica et al. (2021) [21] | Bacteremia | 4357 | EHRs, Microbiological data, Hospital Universitario de Fuenlabrada, Madrid, Spain, (2005–2015) | 51.3% | SVM, RF, KNN | The number of days in ICU before BC extraction, presence of Catheters, Chronic Respiratory disease, Fever, Age, CRP, PLT. |
| | Murri et al. (2024) [22] | HA-BSI | 5660 samples | Generator Center at the Fondazione Policlinico Universitario A. Gemelli IRCCS (FPG), Rome, Italy (2016–2019) | 33.6% | LR | Time BSI > 12 days, Procalcitonin > 1 ng/mL, Presence of a CVC, PLT, Hypotension, BUN, Presence of urinary catheter, Fever, Tachycardia, Altered mental status, Age, Bilirubin, Creatinine |
| Inpatients with SIRS | Ratzinger et al. (2018) [23] | Bacteremia | 466 | EHRs Vienna General Hospital, Austria, (2011–2012) | 28.8% | RF, ANN, ENR | PCT, LBP, Albumin, Bilirubin |
| Inpatients with CVC | Rahmani et al. (2022) [24] | CLABSIs | 27619 | EHRs, a proprietary national longitudinal EHR repository, Houston, Texas, USA (2015–2020) | 1% | XGB, DT, LR | Temperature, HGB, comorbidities, Age, WBC, Race, Neutrophil. |
| | Beeler et al. (2018) [25] | CLABSIs | 70218 | EHRs, Indiana University Health Academic Health Center, USA, (2013–2016) | 0.6% | RF, LR | Age, Gender, history of CLABSI, CHG (Chlorhexidine Gluconate) Bathing Non-compliant Days, Line days. |
| HD patients | Zhou et al. (2023) [26] | Bacteremia | 391 | EHRs, Department of Nephrology, Affiliated Hospital of North Sichuan Medical College, Sichuan Province, China, (2018–2022) | 18.9% | LR, SVM, DT, RF, XGB | PCT, Temperature, Non-arteriovenous fistula dialysis access, NLR, Leukocyte, dialysis duration, LMR, Albumin, Neutrophil, PLT, Age, DBP, CRP, PLR, ALP, SBP, HR, history of BSI, |

*(Continued)*

**Table 1.** (Continued)

| Setting | First author, year | Target condition | No. Patients | Data source | Prevalence | ML models | Key predictors |
|---|---|---|---|---|---|---|---|
| Cancer patients | Yoo et al. (2021) [27] | Candidemia | 34574 | EHRs, academic single hospital in Seoul, Republic of Korea, (2010–2018) | 0.6% | LR, ANN, RF, GBM, AML | Variables reflecting the dynamic status of patients with cancer, including blood urea nitrogen level, 7-day variance of RR, Total bilirubin level, 7-day variance of SBP, Body weight. |
| Maternity patients | Mooney et al. (2020) [28] | Bacteremia | 129 | CBC parameters, Rotunda Hospital, Ireland (2019) | 3% | CART, LDA, KNN, SVM, RF | NLR, CBC parameters. |
| Patients with PCT ≤2.0 ng/ml | Su et al. (2021) [29] | Bacteremia | 931 | EHRs, Mindong Hospital Affiliated to Fujian Medical University, China, (2014–2020) | 47% | ANN, KNN, LR, RF, SVM, and NB. | Interleukin-6, PCT, D-dimer, Lactic acid, Leukocytes, Neutrophil, and PLT. |
| HIV patients | Wu et al. (2023) [30] | Bacteremia | 498 | EHRs, Wenzhou Central Hospital, China, (2014–2021) | 34.3% | SVM, ANN, GBM, GLM, MDA, PLR, NB, RF | Low Hb, CD4+T cell, PLT, LDH, BUN, splenomegaly, absence of ART treatment, Strip shadow, Nodular shadow, and Shock. |
| ED | Choi et al. (2022) [31] | Bacteremia | 24768 | EHRs, An urban tertiary referral hospital, Republic of Korea, (2016–2018) | 12% | XGB, RF, LR | Chief complaint, Age, Temperature, HR, and DBP at triage stage. Neutrophils, PLT, CRP, Chief complaints, and Creatinine at disposition stage. |
| | Choi et al. (2023) [11] | Bacteremia | 15362 | EHRs, Seoul National University Hospital, Seoul National University Bundang Hospital, Republic of Korea, (2016–2018) | 10.9% | BNN | Age, HR, Temperature, DBP, History of chills, Ambulance use |
| | Boerman et al. (2022) [32] | Bacteremia | 4885 | EHRs, Amsterdam UMC, location VU University Medical Center, NL, (2018–2020) | 12.2% | GBT, LR | Bilirubin, Urea, lymphocyte, Pulse rate, CRP, Neutrophil, age, Temperature, DBP, Potassium, Glucose, Thrombocytes, Creatinine, ALP, SBP, Organ damage |
| | Chang et al. (2023) [33] | Bacteremia | 20636 | EHRs, CPD, CBC/DC, China Medical University Hospital, Taiwan, (2021–2022) | 10.4% | CatBoost, LGBM, XGB, RF, LR | Demographics, CPD, CBC/DC |
| | Schinkel et al. (2022) [34] | Bacteremia | 6421 | EHRs, Amsterdam UMC, (VUMC, AMC, ZMC, and BIDMC), NL, (2016–2021) | 5.4% - 12.3% | XGB, LR | Temperature, Creatinine, CRP, Lymphocytes, DBP, Bilirubin, Thrombocytes, Neutrophils, ALP, HR, SBP, Leukocytes, Glucose, Age, Potassium, BUN, Sodium, monocytes |
| ED patients with SIRS | Goh et al. (2022) [35] | Bacteremia | 40395 | EHRs, National Cheng Kung University Hospital, Taiwan, (2015–2019) | 10% | LR, SVM, RF | Age, Gender, COPD, Uncomplicated DM, Hemato-oncology, WBC, Band cell, Platelet, Temperature, HR, mild liver disease, Mean arterial pressure, RR, GSC |
| ED patients with fever | Tsai et al. (2023) [36] | Bacteremia | 3669 | EHRs, Chi Mei Medical Center, Taiwan, (2017–2020) | 13.8% | RF, LR, MLP, XGB, LGBM | Hypertension, Gender, Temperature, DM, Age, CRP, PLT, WBC, Malignancy, Eosinophil, HR, BMI, Hb, RR, SBP, DBP, Band, CKD, Liver Cirrhosis, COPD, GCS |

*(Continued)*

**Table 1.** (Continued)

| Setting | First author, year | Target condition | No. Patients | Data source | Prevalence | ML models | Key predictors |
|---|---|---|---|---|---|---|---|
| ICU patients | Roimi et al. (2020) [37] | Bacteremia | 3372 | EHRs, BIDMC, Boston, Massachusetts, USA, (2008–2012), ICU of Rambam Healthcare Campus (RHCC), Israel, (2013–2017) | ICU acquired: 6.4% (BIDMC), 15.9% (RHCC) | RF, XGB | Time duration (days) between sampling time and last defecation, Time duration (hours) between sampling time and the maximum BUN (mg/dL) value measured during the 5 days prior to sampling, Length of stay (days) between sampling time and ICU admission, The minimal weight (kg) during the 5 days prior to sampling, The time duration between sampling time and the maximum MCHC (g/dL) during the 5 days prior to sampling |
| | Van Steenkiste et al. (2019) [38] | Bacteremia | 2177 | EHRs, ICU, Ghent University Hospital, Belgium, (2013–2015) | 10.5% | BiLSTM, ANN, SVM, KNN, LR | Temperature, Thrombocytes, Leukocytes, CRP, sepsis-related organ failure assessment, HR, RR, PT, and mean systemic arterial pressure. |
| | Boner et al. (2022) [39] | Bacteremia | 6557 | EHRs, ICU, University of Virginia, USA, (2011–2015) | 13.3% | FNN, GRU, CNN, LR | Temperature, BUN, BP, HR, Albumin, PLT, Chloride, Creatinine, Chloride, and Phosphorus. |
| | Pai et al. (2021) [40] | Bacteremia | 4275 | EHRs, Taichung Veterans General Hospital ICU, Taiwan, (2015–2019) | 13.8% | LR, SVM, MLP, RF, XGB | ALP, CVC period, prothrombin time, PLT, Albumin, Apache II score, Age, foley |
| ICU patients with CVC | Parreco et al. (2018) [41] | CLABSIs | 57786 admissions | MIMIC-III database, USA, (2001–2012) | 1.5% | LR, GBT, DL | Severity of illness scores (like SAPS II, APS III, and OASIS) and comorbidities. |
| ICU patients with new-onset SIRS | Yuan et al. (2021) [42] | Candidemia | 7932 | EHRs, Peking Union Medical College Hospital, The Affiliated Hospital of Qingdao University, The First Affiliated Hospital of Fujian Medical University, China, (2013–2017) | 1% | XGB, SVM, RF, ET, LR | Colonization, Diabetes, AKI, total number of parenteral nutrition days, history of fungal infection, CRRT days, Abdominal surgery, BDG, days of mechanical ventilation, Length of hospital and ICU stay, days of CVC |

patients collected between 2016 and 2018. The models utilized demographics, chief complaints, vital signs, and laboratory data collected during ED triage and disposition. Two primary models were developed: the Triage eXtreme Gradient Boosting (XGB) and Disposition XGB models. In a subsequent study, Choi et al. (2023) aimed to refine the predictive accuracy of ED triage-based bacteremia identification using an advanced ensemble of ML techniques. The study analyzed data from over 30,000 ED visits, employing various clinical inputs, including detailed symptom descriptions, vital signs, and initial lab results [11]. The developed model incorporated a GBM framework that effectively integrated the diverse dataset to predict bacteremia risk. The study by Schinkel et al. (2022) harnessed data from EHRs of 44,123 unique ED visits across four hospitals: Amsterdam UMC (VUMC), Zaans Medical Center (ZMC), and Beth Israel Deaconess Medical Center (BIDMC) covering the period from 2011 to 2021 [34]. They employed a hybrid of LR and XGB models, with the latter outperforming in predictive accuracy. The data included demographics, vital signs, and laboratory data such as creatinine and CRP. This predictive model was integrated into the VUMC's EHRs system for real-time prospective evaluation, affirming its practical utility by potentially reducing unnecessary BC analyses by at least 30%. The study by Boerman et al. (2022) utilized a single-center,

retrospective observational design, and the study encompassed data from 51,399 ED visits at the VUMC from September 2018 to June 2020 [32]. Data included demographics, vital signs, laboratory and radiology data, and medications administered during ED visits. The study employed two predictive models: an LR model and a gradient boosted tree (GBT) model. Both demonstrated good predictive performance with an AUROC of 0.77 and 0.78, respectively. Notably, the GBT model was optimized to predict 69% of BC as negative, with a negative predictive value exceeding 94%, indicating its utility in potentially reducing unnecessary BCs and associated healthcare costs. The models harnessed a comprehensive array of features, including commonly available clinical data such as CRP levels and WBC counts, to predict the likelihood of bacteremia. These studies illustrate how integrating ML models into ED workflows can improve the speed and accuracy of BSI detection, potentially reducing unnecessary interventions and optimizing resource allocation. However, all the models focused on current patient data, and none of the studies utilized the predictors from the complete medical history of their patients, apart from demographics and information on co-morbidities. In our previous work, through innovative feature engineering from historical medical records and employing an array of ML classifiers, we showcased the efficacy of tree-based ML models in predicting adverse events in hospitals [44,45].

**1.1.1. Data challenges and strategies.** BSIs may be relatively rare compared to the number of non-infection cases in a dataset. This imbalance can make models biased towards predicting the majority class, reducing their effectiveness in identifying true infection cases. Most studies reported imbalanced datasets with BSI prevalence rates, as in Table 1. To overcome challenges with data imbalance, the Synthetic Minority Over-sampling Technique (SMOTE) was widely used to augment the minority class in the dataset by generating synthetic samples [15,24,42]. Goh et al. (2022) employed oversampling, undersampling, and random oversampling (ROSE) methods for model development [35]. Lien et al. (2022) and Van Steenkiste et al. (2019) employed the Area Under Precision-Recall Curve (AUPRC) metric for a more accurate assessment of model performance in imbalanced datasets [19,38]. The study by Garnica et al. (2021) encountered significant issues with missing data across the patient records used [21]. The types of missing data were classified into three categories: Missing Completely At Random (MCAR), Missing At Random (MAR), and Missing Not At Random (MNAR). They employed a separate class method to represent the missing data, ensuring the ML models could handle these cases without dropping significant data. Using patient data to train ML models can raise privacy and security concerns. Ensuring patient anonymity and complying with regulations can limit the accessibility and use of certain data. Boerman et al. (2022) faced difficulty with the limitation of not being able to use free-text data such as physician and nurse reports due to privacy concerns [32]. To ensure patient privacy and compliance with data protection regulations, researchers can implement effective deidentification of patient records, involving eliminating or altering direct identifiers, such as names, ages, gender, or location, which could be combined to identify an individual.

## 2. Results

### 2.1. Patient characteristics

The dataset's mean patient age was 63.6 years, with a near-equal gender distribution (47.4% female, 52.5% male). To provide a detailed description of the study population, we include Table 2, summarizing the patients' demographics and key clinical characteristics included in the analysis. These characteristics are based on the entire dataset encompassing all care episodes recorded from 1999 to 2020, thus providing a comprehensive overview of the patient population and their interactions with the healthcare system. There was a total of 72,495 BC

**Table 2. Summary of patient and care episode characteristics (1999–2020).**

| Characteristics | Value |
| --- | --- |
| Total number of patients | 35,591 |
| Age (mean) | 63.6 |
| Sex (Male%/Female%) | 52.5/47.4 |
| Median Length of Stays (days) | 0.41 |
| ICU Admissions (%) | 66.2 |
| 30-Day Mortality (%) | 20.8 |
| 30-Day Readmission (%) | 10.4 |
| Number of Blood Cultures | 72,495 |

episodes in the dataset. Following the exclusion of pediatrics and outpatient BC episodes, 65,975 adult inpatient BC episodes were included in the analysis. Of the BC episodes, 5,288 (8%) were classified as positive. Please see the flow chart provided in Fig 1. The differences in the mean values for all features across the two classes and the T-statistic and p-value are given

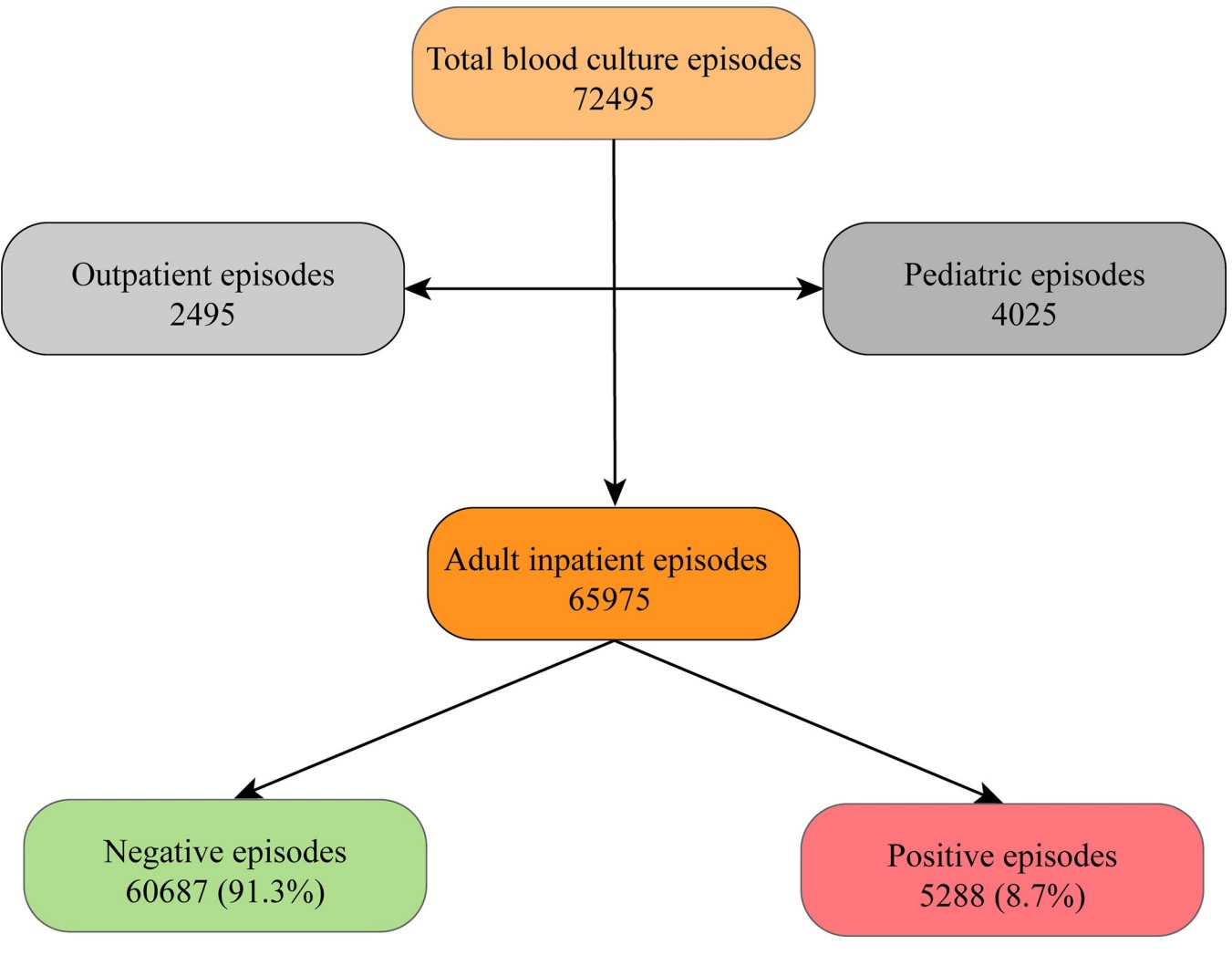

**Fig 1. Flowchart The flowchart depicts the categorization of the BC episodes.**

in (S1 File). The top 25 most significant features, the comparison between their mean values across the two classes, and the T-statistic and p-value are given in (S1 File). There were significant differences between the positive and negative BC groups regarding the occurrence of ICD codes starting with A, B, and N in the medical records. The ICD Chapter I (A00-B99) describes certain infectious and parasitic diseases, and ICD Chapter XIV (N00-N99) concerns kidney and urinary disorders. From laboratory data, bilirubin, creatinine, and CRP showed the most significant differences between the groups. The positive BC group also had higher mean age, higher total length of stay (LOS) till the time of BC, shorter time to the most recent medical episode, and a higher number of previous positive BC test results. The correlation matrix heatmap visually represents the strength and directionality of correlations between various clinical features, highlighting the complex interdependencies relevant to BSI prediction, please see (S1 File). The top three most correlated coefficients among significant features and the list of all the features, their description, and the mean value across the dataset are given in (S1 File).

## 2.2. Model performance

The comparative performance metrics for the sequential and static models in the main and case studies are given in Tables 3 and 4, respectively.

**2.2.1. Sequential model performance.** In the main study, the GRU model achieved the highest AUROC (0.7830) and F1 score (0.3267) among the sequential models. The GRU's relatively high recall (0.6400) compared to its precision (0.2193) suggests that it was particularly sensitive to detecting true positive cases, though at the cost of misclassifying some negative cases as positive. The LSTM model, with an AUROC of 0.7568, also performed reasonably well, although it did not reach the same levels of discrimination as the GRU. On the other hand, the Transformer model, which achieved an AUROC of 0.7643 and an AUPRC of 0.2911, demonstrated moderate performance with a more balanced trade-off between precision (0.2339) and recall (0.5420). CNN-LSTM, with an AUROC of 0.7600 and AUPRC of 0.3115, showed relatively low precision (0.1732) but higher recall (0.6785), indicating a significant tendency to predict positive cases, even at the risk of false positives. This model's specificity (0.7105) was the lowest among all sequential models, reflecting a higher rate of false positives. DKN, as a specialized model designed to incorporate knowledge-aware features, demonstrated

**Table 3. Comparative performance metrics of ML models.**

| Main Study | AUROC | Accuracy | Precision | Recall | F1 Score | AUPRC | Specificity |
|---|---|---|---|---|---|---|---|
| *Sequential Models* | | | | | | | |
| **LSTM** | 0.7568 | 0.7812 | 0.2008 | 0.5592 | 0.2955 | 0.3186 | 0.8011 |
| **GRU** | 0.7830 | 0.7835 | 0.2193 | 0.6400 | 0.3267 | 0.3560 | 0.7964 |
| **CNN-LSTM** | 0.7600 | 0.7079 | 0.1732 | 0.6785 | 0.2760 | 0.3115 | 0.7105 |
| **CNN-GRU** | 0.6973 | 0.8612 | 0.2425 | 0.3256 | 0.2779 | 0.2135 | 0.9091 |
| **Transformer** | 0.7643 | 0.8167 | 0.2339 | 0.5420 | 0.3267 | 0.2911 | 0.8413 |
| **DKN** | 0.6911 | 0.9012 | 0.3412 | 0.2194 | 0.2671 | 0.6000 | 0.9621 |
| **CapMatch** | 0.5003 | 0.0824 | 0.0821 | 1.0000 | 0.1517 | 0.5002 | 0.0004 |
| *Static Models* | | | | | | | |
| **XGBoost** | 0.7995 | 0.8521 | 0.3191 | 0.5531 | 0.4047 | 0.4336 | 0.8876 |
| **LightGBM** | 0.8144 | 0.8046 | 0.2659 | 0.6529 | 0.3779 | 0.4319 | 0.8198 |
| **CatBoost** | 0.8181 | 0.8481 | 0.3219 | 0.6061 | 0.4205 | 0.4490 | 0.8750 |
| **NN** | 0.7739 | 0.9204 | 0.5241 | 0.3141 | 0.3928 | 0.3944 | 0.9745 |
| **LR** | 0.7771 | 0.7497 | 0.2150 | 0.6610 | 0.3244 | 0.3154 | 0.7586 |
| **RF** | 0.8407 | 0.9258 | 0.8000 | 0.1276 | 0.2201 | 0.4677 | 0.9971 |

**Table 4. Case Study: Comparative performance metrics of ML models.**

| Case Study | AUROC | Accuracy | Precision | Recall | F1 Score | AUPRC | Specificity |
|---|---|---|---|---|---|---|---|
| *Sequential Models* | | | | | | | |
| **LSTM** | 0.5919 | 0.5700 | 0.5607 | 0.6061 | 0.5825 | 0.5987 | 0.5347 |
| **GRU** | 0.5927 | 0.5550 | 0.5446 | 0.6162 | 0.5782 | 0.5699 | 0.4950 |
| **CNN-LSTM** | 0.6416 | 0.5600 | 0.5390 | 0.7677 | 0.6333 | 0.6344 | 0.3564 |
| **CNN-GRU** | 0.6624 | 0.6300 | 0.6437 | 0.5657 | 0.6022 | 0.6242 | 0.6931 |
| **Transformer** | 0.6050 | 0.5900 | 0.6545 | 0.3636 | 0.4675 | 0.6294 | 0.8119 |
| **DKN** | 0.6392 | 0.6250 | 0.6277 | 0.5960 | 0.6114 | 0.6295 | 0.6535 |
| **CapMatch** | 0.5500 | 0.5450 | 0.7857 | 0.1111 | 0.1947 | 0.5371 | 0.9703 |
| *Static Models* | | | | | | | |
| **XGBoost** | 0.7702 | 0.6850 | 0.7045 | 0.6263 | 0.6631 | 0.7816 | 0.7426 |
| **LightGBM** | 0.7667 | 0.6700 | 0.6813 | 0.6263 | 0.6526 | 0.7737 | 0.7129 |
| **CatBoost** | 0.8200 | 0.7150 | 0.7283 | 0.6768 | 0.7016 | 0.8308 | 0.7525 |
| **NN** | 0.7820 | 0.7200 | 0.7312 | 0.6869 | 0.7083 | 0.7604 | 0.7525 |
| **LR** | 0.7716 | 0.6800 | 0.6733 | 0.6869 | 0.6800 | 0.7584 | 0.6733 |
| **RF** | 0.7561 | 0.6700 | 0.6941 | 0.5960 | 0.6413 | 0.7615 | 0.7426 |

the highest specificity (0.9621), suggesting its strength in ruling out non-BSI cases with high confidence. However, its lower recall (0.2194) and precision (0.3412) indicates that while it effectively identified true negatives, it struggled with true positives, leading to a trade-off that limits its overall utility in a clinical prediction setting where sensitivity and specificity are crucial. The high recall (1.0000) of CapMatch is deceptive, as it essentially labeled all cases as positive, which, while capturing all true positives, resulted in an overwhelming number of false positives, rendering the model impractical for actual use. In summary, the analysis of these sequential models highlights significant trade-offs between sensitivity/recall and specificity. Models like GRU and Transformer balanced these trade-offs better than others, making them more suitable for clinical applications where false positives and negatives can have serious consequences. However, even the best-performing sequential models faced challenges, particularly in dealing with the inherent class imbalance, which affected their precision and overall reliability.

**2.2.2. Static model performance.** The static models generally outperformed the sequential models across most evaluation metrics, indicating their stronger suitability for the BSI prediction task on this dataset. The RF model stood out with the highest AUROC (0.8407) and an exceptional specificity (0.9971), underscoring its effectiveness in correctly identifying true negative cases, those patients not likely to develop BSI. This high specificity suggests that RF is particularly adept at minimizing false positives, which is crucial in clinical settings where unnecessary treatments can have serious consequences. However, the RF model's recall was notably low (0.1276), indicating a significant trade-off. While the model was extremely reliable in ruling out non-cases, it failed to identify a substantial portion of actual BSI cases, reflected in its lower recall. The CatBoost, a tree-based ensemble model, demonstrated robust performance across most metrics, with an AUROC of 0.8181, indicating a strong overall discrimination ability. It achieved a balanced F1 score of 0.4205, suggesting a more moderate trade-off between precision and recall compared to the RF model. CatBoost's recall (0.6061) was significantly higher than that of RF, indicating that it was more effective in identifying true positives, albeit with a slightly lower specificity (0.8750). XGBoost, another popular gradient-boosting model, also performed well with an AUROC of 0.7995, reflecting its capability to differentiate between positive and negative cases. Its precision (0.3191) and recall (0.5531) were relatively

balanced, leading to a moderate F1 score of 0.4047. This indicates that XGBoost, while slightly less powerful than CatBoost, provided a good trade-off between precision and recall, making it a reliable choice for BSI prediction. Its specificity (0.8876) was slightly lower than CatBoost's, suggesting that while XGBoost was effective, it might be slightly more prone to false positives compared to CatBoost. With an AUROC of 0.8144, LightGBM closely followed CatBoost in overall performance. Its recall (0.6529) was among the highest, indicating its strong sensitivity to true positives. However, this came with a trade-off in specificity (0.8198), which, while still high, was lower than that of RF and CatBoost, indicating a slightly higher rate of false positives. This suggests that LightGBM might be more suitable in scenarios where the cost of missing a positive case is higher than that of a false positive. The NN model also performed well, with an AUROC of 0.7739 and the highest precision (0.5241) among all models, reflecting its ability to minimize false positives. However, its recall (0.3141) was lower, leading to an F1 score of 0.3928. This suggests that the NN model was particularly conservative in its predictions, favoring precision over recall, which could be advantageous in situations where the cost of a false positive is high. The NN's high specificity (0.9745) reinforces this interpretation, indicating that it was very effective at identifying true negatives, making it a reliable choice for tasks where precision is paramount. LR, the simplest model in the static models set, also showed competitive performance with an AUROC of 0.7771. However, its precision (0.2150) was the lowest among the static models. Its recall (0.6610), however, was the highest, indicating that it was effective in identifying true positives, though this came at the expense of a higher rate of false positives. The LR model's F1 score (0.3244) and specificity (0.7586) were also lower, indicating that while it performed adequately, it was outperformed by the more sophisticated ensemble methods like CatBoost and LightGBM. In summary, the static models generally performed better than the sequential models, with tree-based ensembles like RF, CatBoost, and XGBoost leading the way. While RF excelled in specificity, models like CatBoost and LightGBM provided more balanced performance across different metrics, making them versatile tools for clinical prediction tasks.

**2.2.3. Case study: Model performance.** In the case study conducted to further validate the models on a balanced subset of the data, there were some notable shifts in performance, reflecting the variability in model generalization. Among the sequential models, the CNN-GRU model performed the best, achieving an AUROC of 0.6624 and an AUPRC of 0.6242. The balanced dataset allowed CNN-GRU to maintain a relatively high precision (0.6437) and F1 score (0.6022), which indicates a good trade-off between precision and recall. DKN model also showed good performance among the sequential models, with an AUROC of 0.6392 and a balanced AUPRC of 0.6295. The static models once again outperformed the sequential models, reaffirming their robustness and reliability in the context of BSI prediction, even when the data distribution is balanced. The CatBoost model stood out, achieving the highest AUROC (0.8200) and AUPRC (0.8308) across all models. These metrics indicate that CatBoost maintained its strong discriminative power and excelled in precision-recall performance, making it highly effective at distinguishing between true positives and negatives. The high AUPRC is significant as it reflects the model's ability to accurately predict the minority class (BSI-positive cases) even in a balanced setting, where the opportunity to achieve a high AUPRC is more challenging due to the increased presence of true negatives. The NN model achieved an AUROC of 0.7820 and an AUPRC of 0.7604. In terms of precision (0.7312) and recall (0.6869), the NN model demonstrated a well-balanced trade-off, resulting in a solid F1 score of 0.7083. This balance suggests that the NN was particularly effective in identifying true positives while maintaining a relatively low rate of false positives. This is a critical advantage in medical prediction tasks, where the consequences of both false negatives and false positives can be significant. The NN model's high specificity (0.7525) further highlights its ability to

identify true negatives correctly. In summary, the case study results, conducted on a balanced dataset, reinforced the findings from the main study by demonstrating that static models are more reliable and effective. Among the sequential models, CNN-GRU showed the best performance, followed closely by DKN, indicating their potential utility in scenarios where leveraging both temporal dynamics and external knowledge is beneficial.

## 2.3. Global feature importances

The global feature importance of the models was assessed using SHAP values, providing insights into the contributions of each feature to the model's predictions. SHAP values offer a unified framework to interpret the output of ML models, primarily designed for tree-based models like the XGB used in our study. The bar plots rank the features by their average impact on model output magnitude, giving a clear picture of which variables are most significant in the decision-making process. The dot plots, or beeswarm plots, provide a more nuanced view by showing how the values of these features (colored by feature value) contribute to the prediction. Positive SHAP values indicate that a feature contributes positively towards the prediction, while negative SHAP values indicate that a feature contributes negatively.

In the main study (Fig 2), the SHAP summary bar plot indicates that the most influential features for predicting BSI include the count of diagnostic codes starting with letters A, B, and N, bilirubin levels (*BILIRUBIN_TOTAL*), creatinine levels (*KREATININ*), and leukocyte count (*LEUKOCYTTER*). These features are followed by the age of the patient, the number of previous positive BC results, and the length of stay (LOS) during the current or recent hospital episode. The SHAP summary dot plot further clarifies the impact of each feature's value on the model's output. For instance, higher bilirubin, creatinine, and leukocyte values (as depicted by red dots) are associated with a higher probability of a positive BSI prediction, reflecting their clinical significance as indicators of infection severity. Conversely, lower thrombocyte levels (*TROMBOCYTTER*) indicate a higher risk of BSI. The case study (Fig 3) also reveals critical

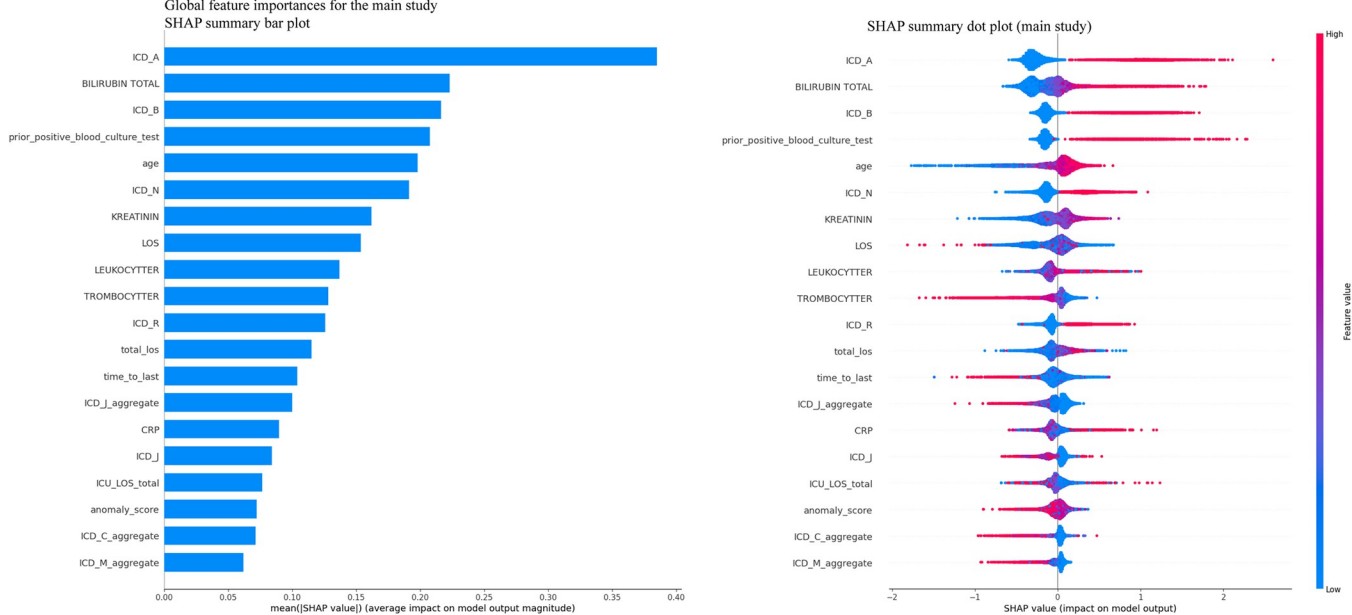

**Fig 2. Main study: SHAP summary plots for XGB model.** The bar plot on the left illustrates the global feature importance ranked by the sum of SHAP values across all samples. On the right is the Beeswarm plot detailing the individual SHAP values for each feature and their impact on the model's output.

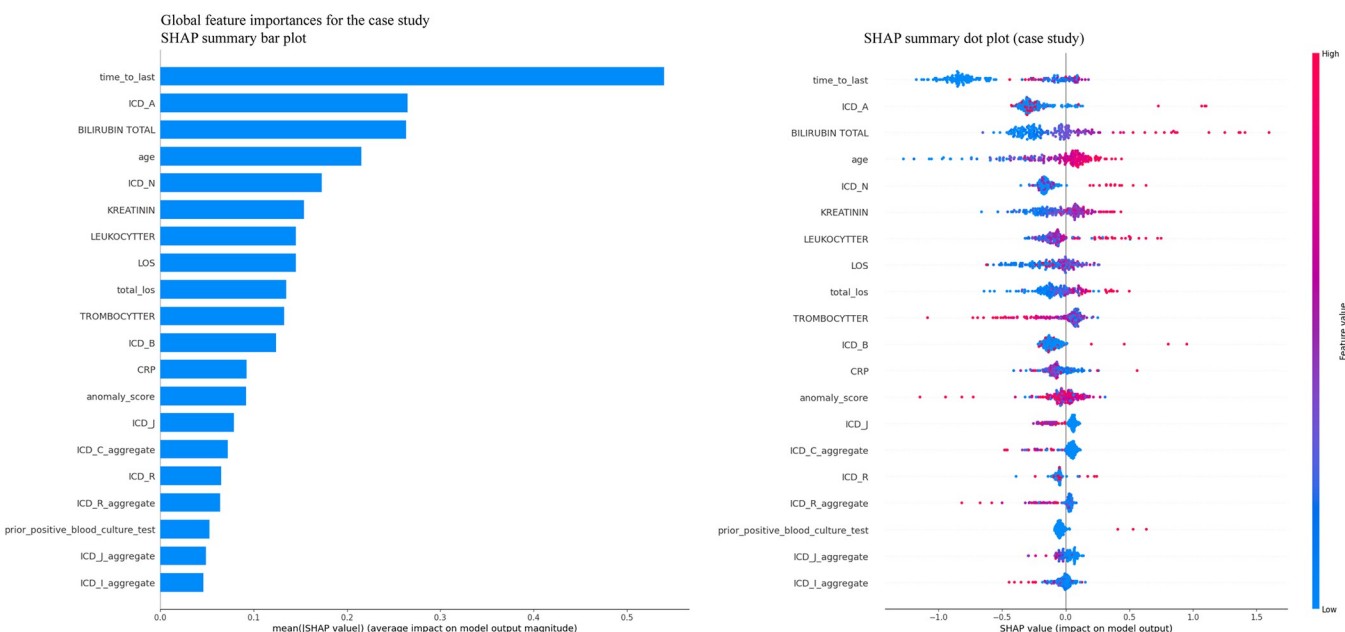

**Fig 3. Case study: SHAP summary plots for XGB model.** The bar plot on the left illustrates the global feature importance ranked by the sum of SHAP values across all samples. On the right is the Beeswarm plot detailing the individual SHAP values for each feature and their impact on the model's output.

differences in global feature importance compared to the main study. The SHAP summary bar plot for the case study highlights "time to last" (the time to the most recent episode) as the most significant predictor, followed by the count of diagnostic codes starting with letters A, bilirubin levels, and age. This suggests that in the case study, the recency of patient interactions with the healthcare system is a more prominent predictor of BSI risk. The SHAP summary dot plot provides further details, showing that shorter times to the most recent episode (indicated by the higher concentration of red dots on the left) are strongly predictive of positive BSI outcomes. Additionally, the importance of diagnostic codes starting with letters A and total bilirubin levels remains consistent with the main study, emphasizing their relevance across different contexts.

## 2.4. Local feature importances

Fig 4. details the waterfall and force plots for the first three prediction tasks. In the first prediction task, the feature *ICD_A* (infectious and parasitic diseases) has a significantly high positive SHAP value, indicating that an increase in counts of ICD-10 Chapter I codes in the medical history strongly sways the model towards predicting BSI. Similarly, creatinine level and *ICD_N* (genitourinary diseases) show negative SHAP values, implying that higher levels of creatinine and the presence of genitourinary diseases are linked to a lower probability of BSI in this instance. For the second prediction, counts of prior positive BC results have the most substantial positive impact, aligning with clinical reasoning that past positive tests could indicate a higher risk for infection. *ICD_B* (infectious and parasitic diseases) and *ICD_R* (abnormal symptoms and findings) have a positive effect, and total bilirubin has a positive contribution, predicting a higher risk of BSI. Similar to the second prediction, counts of prior positive BC results again show a positive influence in the third prediction. Additionally, '*ICD_C_aggregate*' (aggregate count of cancer-related codes) exerts a negative impact on the prediction outcome, whereas '*procedure_W_aggregate*' (count of procedures on female reproductive organs in the

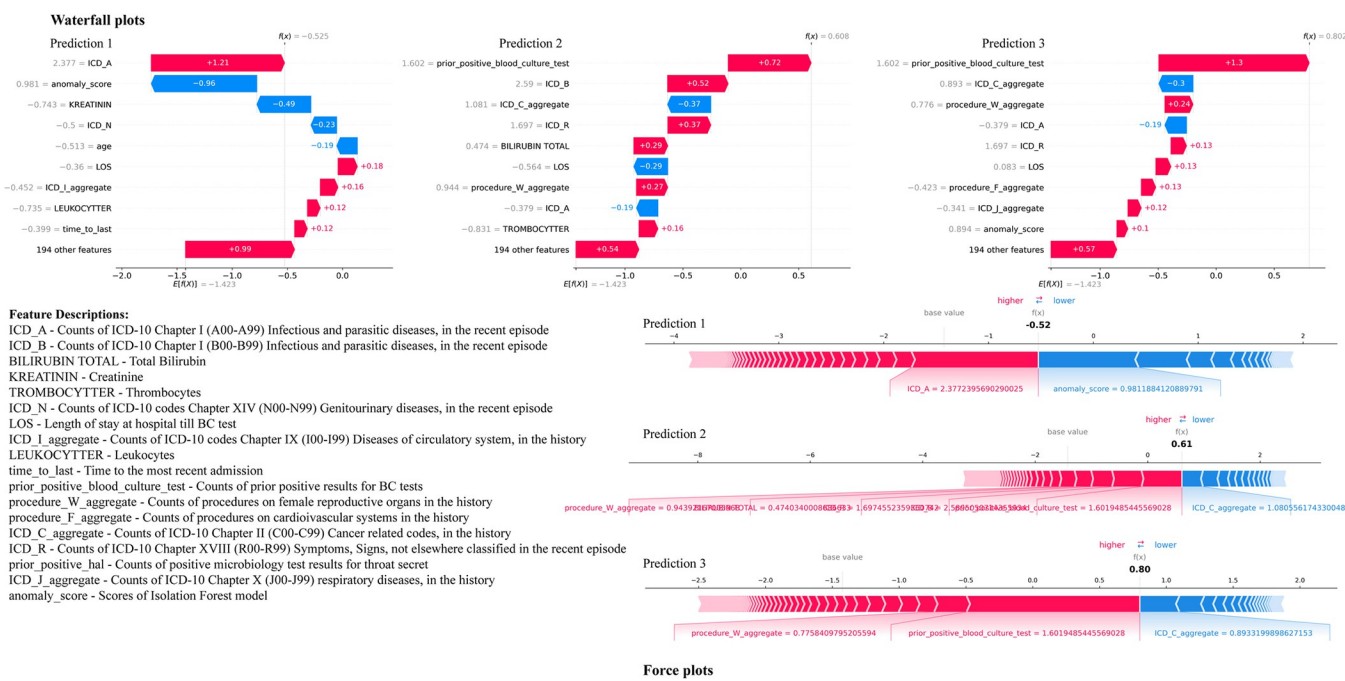

**Fig 4. Waterfall and Force plots for the first three predictions from the test set.**

history) is associated with an increase in the risk of BSI. In our model, the 'anomaly_score' feature was observed to have a negative SHAP value for many predictions. This should not be misconstrued as an indication of low importance. On the contrary, the negative SHAP value reflects that higher values of 'anomaly_score' reduce the probability of predicting a BSI. This behavior is crucial for the model's performance, as it highlights the features that drive the decision away from a positive prediction, thereby improving the model's specificity.

## 2.5. Results of statistical analysis

The statistical analysis revealed several significant differences between the BSI positive and negative groups, highlighting critical predictors, also referred to as features throughout this study. The top 25 most significant features are detailed in (Table A in S2 File), where features such as *ICD_A* (infectious diseases), *ICD_B* (infectious diseases), and *ICD_N* (kidney disease) demonstrated the most substantial differences between the groups, with extremely low p-values (e.g., *ICD_A*: $p < 1e\text{-}30$), indicating their strong association with BSI outcomes. Specifically, the mean bilirubin level was significantly higher in the BSI-positive group (14.65 vs. 7.32; $T = -23.83$, $p < 1e\text{-}100$), suggesting its potential as a biomarker for BSI prediction. Similarly, prior positive BC tests showed a notable difference (0.37 vs. 0.18; $T = -20.33$, $p < 1e\text{-}90$), underscoring the importance of historical infection data in predicting future BSI episodes. (Table B in S2 File) expands on these findings by providing a detailed breakdown of the statistical significance of each clinical feature. The analysis also included the construction of a Pearson correlation matrix (Fig A in S2 File), which highlighted the complex interdependencies among the clinical features relevant to BSI prediction. For instance, there was a strong positive correlation between creatinine levels and kidney-related ICD codes, which further supports the role of renal function in the risk of BSI. In our analysis, we observed that certain features exhibit significant correlations, as detailed in (Table C in S2 File). These correlations could potentially influence the SHAP values by making it challenging to isolate the individual

contribution of each feature. In the context of tree-based models like XGBoost, SHAP values assume feature independence when attributing importance to each feature. However, when features are highly correlated, SHAP values may reflect shared information between these features rather than their distinct contributions, leading to inflated or distorted importance scores. Therefore, we interpreted SHAP values cautiously, especially when correlated features might skew the feature ranking.

## 3. Discussions

In this study, we presented the XBSI framework for early prediction of BSI in hospitals. The results indicate that the AUROC ranged from 0.5003 to 0.7830 for the sequential models. In contrast, the static models showed superior AUROC values ranging from 0.7771 to 0.8407. Additionally, the case study on a balanced subset of the data reinforced these findings, demonstrating that the CNN-GRU model performed the best among sequential models. In contrast, the CatBoost model outperformed all models with the highest AUROC of 0.8200. The statistical analysis and global and local SHAP value interpretations reveal that the count of diagnostic and procedural ICD codes and laboratory data significantly contribute to the model's predictions. These findings align with existing literature, underscoring the clinical relevance of laboratory data and medical history for early BSI prediction [14–16,20,21,46]. The demonstrated efficacy of our XBSI framework highlights the predictive capability of combining medical history with laboratory data for early BSI prediction at the time of the BC order. The modular design of our framework reinforces the robustness to all types of medical data and emphasizes the utility of integrating diagnostic and procedural ICD codes into predictive models. This approach allows for a nuanced understanding of patient profiles, which is pivotal for effectively implementing predictive healthcare solutions. Our method demonstrates several distinct advantages over SOTA models like DKN and CapMatch, primarily due to our unique feature engineering and model interpretability strategies. While the DKN model integrates domain knowledge through a knowledge-aware attention mechanism, enabling it to utilize external information for enhancing predictive performance, it often sacrifices model interpretability. Moreover, our feature engineering process distinguishes our method by extracting and selecting features from the raw EHR data that are particularly relevant to BSI prediction. These features, combined with the interpretability provided by SHAP values, ensure that our model is not only accurate but also comprehensible to clinicians.

### 3.1. Interpretations of results: Performance of sequential vs. static models

The sequential models were designed to capture temporal dependencies by incorporating sequence information. In contrast, the static models, did not explicitly account for temporal sequences but instead relied on feature engineering to encapsulate time-dependent information. To ensure that temporal patterns were not entirely overlooked by the static models, we derived specific temporal features, such as 'time_to_last' episode (the time elapsed since the last episode) and 'total_los' (the cumulative hospital LOS up to the BC), which were included as input features. These engineered features allowed the static models to incorporate some temporal aspects indirectly. Despite our initial hypothesis that the sequential models would outperform the static models due to their ability to directly process time-dependent data, the results demonstrated that static models achieved better overall performance. Several factors may contribute to this observation:

 **3.1.1. Data characteristics and temporal dependencies.** The temporal patterns within the dataset may not have been sufficiently strong or consistent to provide a distinct advantage to the sequential models. Static models, by focusing on optimizing feature importance without

the complexity of sequence modeling, may have been better suited to the nature of our data. The static models benefited greatly from the feature engineering process, where temporal aspects were distilled into features like 'time_to_last' episode and 'total_los' (cumulative LOS till the BC).

**3.1.2. Model complexity and data volume.** Sequential models generally require larger datasets to effectively capture subtle temporal patterns. Given the size and characteristics of our dataset, the sequential models might have been limited in their ability to generalize, resulting in lower performance compared to the static models. Given the relatively smaller sizes and irregularities of sequences of medical events, the complexity of these models may have led to overfitting or underperformance. In contrast, the static models, which rely on simpler assumptions about the data, were able to generalize better under these conditions.

**3.1.3. Data enchancement.** The static models benefited from comprehensive data enhancement through feature engineering, which distilled critical temporal information into specific features. This process likely captured much of the necessary temporal dependencies, allowing the static models to perform effectively. These findings suggest that while sequential models are theoretically well-suited for tasks involving temporal data, their practical performance may be constrained by the dataset size, the strength of the temporal patterns, and the effectiveness of feature engineering. Further research is needed to explore the specific conditions under which sequential models may outperform static models in this context.

**3.1.4. Interpretability and clinical applicability.** The static models also provided clearer and more interpretable results, which is a critical consideration in clinical settings. The SHAP value analysis conducted on the XGBoost model, for example, offered insights into feature importance that were straightforward for clinical practitioners to understand and act upon. This level of interpretability is essential for integrating AI-based models into healthcare decision-making processes.

**3.1.5. Future research directions.** While the findings suggest that static models are well-suited for the current prediction task, it is important to note that this may not universally apply to all clinical prediction tasks or datasets. Future research could explore conditions under which sequential models might outperform static ones, particularly in larger datasets or where temporal patterns are more pronounced. Additionally, hybrid approaches that combine the strengths of both static and sequential models could offer a balanced solution, leveraging the interpretability of static models and the temporal awareness of sequential models.

## 3.2. Comparison with previous works on BSI prediction models

Comparing the recent studies on predicting BSI in hospital settings, Bhavani et al. (2020) utilized LR and GBM models to predict bacteremia and fungemia from EHRs, achieving AUCs of 0.73 and 0.88, respectively. The key predictors identified in the study were time from admission to BC, temperature, age, heart rate, prior bacteremia/fungemia, WBC, blood urea nitrogen (BUN), glucose, diastolic blood pressure (DBP), and systolic blood pressure (SBP) [14]. Lee et al. (2019) compared multiple ML algorithms, including MLP, SVM, and RF, achieving their best AUC of 0.732 with the RF model [15]. The key predictors reported by the study included alkaline phosphate (ALP), platelet, maximum body temperature, SBP, WBC, CRP, ICU stay, hospital day-to-BC, age, heart rate, prothrombin time, and albumin. In a subsequent study, Lee et al. (2022) further explored MLP, RF, and XGB, focusing on their application over a long-term dataset. Their models achieved AUCs of 0.762 for MLP and 0.758 for RF in the 12-hour data group, slightly lower than our CatBoost model's performance. The key predictors in this study were monocyte, platelet, hospital stay, neutrophil, total bilirubin, BUN, albumin, ALP, WBC, CRP, creatinine, pulse rate, and chloride [16]. The study by Mahmoud et al. used

data from a tertiary care center comprising patient demographics, LOS before BC collection, presence of central line, vital signs, laboratory data, and SIRS and qSOFA scores [20]. They employed various models, including NN and LR, with their best models achieving modest performance metrics (highest specificity at 89% but with low sensitivity). This study extensively used vital signs and other real-time clinical parameters such as temperature and heart rate as predictors. Despite this, the highest sensitivity achieved was only 31% with LR, and even though some of their models achieved high specificity, they needed more sensitivity, limiting their practical utility in clinical settings. Our model's ability to outperform these studies without needing immediate clinical data or vital signs showcases our approach's robustness and efficiency. It suggests a potential for earlier and simpler implementation in clinical workflows. This is particularly advantageous in healthcare settings where immediate comprehensive data collection is challenging, offering a powerful tool for early BSI prediction that is less dependent on the specific timing of clinical data acquisition. Similar to our study, the study by Garnica et al. (2021) used SVM, RF, and KNN, with a combination of RF and SVM yielding the best performance metrics [21]. The number of days in ICU before BC extraction, presence of catheters, age, chronic respiratory disease, fever, and CRP were reported as key predictors. Their better performance in comparison can also be attributed to the fact that they used a BSI dataset with a prevalence rate of 51.3%, the highest reported among all the studies reporting ML-based BSI prediction models.

Our study's findings also contribute to the evolving landscape of ML applications for BSI prediction in ED settings. Schinkel et al. employed LR and XGBoost models using vital signs, laboratory data, and demographics [34]. Their XGBoost model achieved an AUROC of 0.81 (95% CI 0.78–0.83), slightly higher compared to our XGBoost model AUROC of 0.7951 (95% CI 0.7833–0.8069) and an AUPRC of 0.34 (95% CI 0.29–0.38) slightly lower than our XGBoost model AUPRC of 0.3942. The key predictors brought out by this study were temperature, creatinine, CRP, lymphocytes, DBP, bilirubin, thrombocytes, neutrophils, APL, heart rate, SBP, leukocytes, glucose, age, potassium, BUN, sodium, monocytes. Their model effectively reduced unnecessary BC by approximately 30% during real-time prospective evaluation, which aligns with our goals of enhancing diagnostic efficiency and reducing healthcare costs. Our study builds on this foundation by implementing a similar ML approach. Still, it extends its application by incorporating a more comprehensive array of clinical variables derived from historical EHRs and employing a novel algorithmic configuration that may provide improved predictive performance. Similar to our study, Boerman et al. focused on the ED setting of a large teaching hospital, developing predictive models specifically for BSI outcomes based on data available at the end of ED visits, such as demographics, vital signs, administered medications, and laboratory data. They reported an AUC of 0.77 for the GBT model and 0.78 for the LR model, indicating good performance in predicting bacteremia in ED [32]. The key predictors in their study were bilirubin, urea, lymphocytes, pulse rate, CRP, neutrophil, age, temperature, DBP, potassium, glucose, thrombocytes, creatinine, ALP, SBP, and organ damage. The study highlighted the ability of their models to significantly reduce unnecessary BC by predicting negative outcomes with a high degree of accuracy, reflected by a negative predictive value of over 94%. In parallel, our model confirms these findings and demonstrates improved predictive efficiency in differentiating between positive and negative BSI outcomes, which could further optimize the use of resources in ED settings. In another study, Choi et al. (2022) demonstrated the XGB model's effectiveness in predicting bacteremia at different stages of patient care in the ED, achieving an AUROC up to 0.853 [31]. The key predictors in the study were chief complaint, age, temperature, heart rate, and DBP at the triage stage, as well as neutrophils, platelets, CRP, chief complaint, and creatinine at the disposition stage. Their phased approach using predictions at both triage and disposition stages aligns with our methodology

of employing dynamic modeling to adapt predictions based on real-time data updates. Our model's ability to accommodate historical clinical variables may explain any improvements in prediction robustness compared to the framework used by Choi et al. (2023) [11]. The key predictors in this study were age, vital signs, history of chills, and ambulance use. The collective insights from these comparisons suggest that while our model shares common ground with existing approaches, it also explores additional layers of complexity, such as patient recent hospital interactions, comorbidities, and previous history of infections, which may influence the generalizability and effectiveness of the model across diverse healthcare environments, capturing more subtle nuances of BSI risk factors early and could be pivotal in reducing BSI misdiagnosis.

Our framework's modular design facilitates the inclusion of additional data types as they become available, enhancing its adaptability across various clinical settings. This feature is particularly valuable in healthcare environments where flexibility and comprehensive data utilization are crucial for advancing diagnostic accuracy. Moreover, the intuitive nature of the XBSI framework ensures that it can be seamlessly integrated into existing clinical workflows, making it a practical tool for clinicians seeking to leverage AI for improved patient outcomes. The principal clinical value of our approach lies in the ability to identify patients at low risk of a positive BC at the time of suspicion of BSI without the need for waiting to capture vital signs at the moment or within a specified time window, which could increase patient risk and stay [47,48]. Integrating our proposed hospital framework as a pre-emptive BSI prediction tool can reduce BC ordering and its resulting costs and harms [2]. As reported in the literature, the use of data not routinely captured in clinical practice is the main reason why none of the prediction models have been implemented in clinical practice yet [49]. Moreover, in the context of ICU patient monitoring, the application of AI to real-time data may seem redundant. Patients in critical care are already under intensive surveillance, and using AI for immediate alert systems could potentially clutter the workflow rather than enhance it. Instead, a more strategic use of AI lies in its ability to predict a patient's worsening condition well before critical thresholds are reached. By analyzing historical EHRs, AI can identify subtle patterns and indicators of decline that precede acute episodes, thereby enabling preemptive medical interventions. Moreover, as the complexity of predicting health events increases the earlier the prediction occurs, achieving perfect accuracy, as often highlighted through metrics like AUROC in ICU settings [37–42], may not be as critical as maintaining reasonable predictability at the initial stages of patient contact. This approach could shift the focus from crisis response to proactive patient care management, optimizing outcomes through early and targeted intervention. The only limitation of this study is its reliance on data from a single center, which may not represent the diverse patient demographics. Despite these constraints, our model's ability to integrate a broader spectrum of data and apply a XAI-based framework supports its potential utility in clinical settings, promising reductions in unnecessary interventions and improvements in patient management. Furthermore, using historical patient data and various ML techniques allowed for a more flexible and scalable prediction framework. This aspect of our model enhances its applicability and significantly advances BSI prediction using AI.

### 3.3. Addressing critical research gaps in BSI prediction

In the current landscape of BSI prediction research, several critical gaps have hindered the development of effective and reliable models. Our study has been designed to address these challenges, bringing forth significant improvements in data handling, feature engineering, and model interpretability.

**3.3.1. Data imbalance.** One of the predominant issues in existing BSI prediction models is the challenge posed by imbalanced datasets, where minority class instances (e.g., true positive BSI cases) are significantly underrepresented. This imbalance often leads to biased models that perform well on the majority class but fail to accurately identify critical minority class cases, which are crucial in clinical settings. To address this, our study implemented class weighting techniques and conducted a focused case study on a balanced dataset. By adjusting the class weights and balancing the dataset, we significantly enhanced the model's sensitivity and its ability to identify minority class instances, thereby improving the overall performance and clinical applicability of the model.

**3.3.2. Feature engineering.** Another gap in previous studies is the reliance on predefined or less comprehensive feature sets, which can limit the predictive power of the models. In contrast, our approach emphasizes the extraction of meaningful features directly from historical EHRs. This process involves deriving features that are specifically predictive of BSI, rather than relying solely on predefined features. By focusing on comprehensive feature engineering, we have enhanced the model's ability to make accurate predictions, thereby setting our approach apart from existing methodologies.

**3.3.3. Model interpretability.** A major limitation in many SOTA models is their lack of transparency, which poses a barrier to their adoption in clinical practice. Models that operate as "black boxes" are often met with skepticism by healthcare professionals who require clear, interpretable insights into how predictions are made [50]. To overcome this, we have employed SHAP values in our model, providing a transparent view of the contribution of each feature to the model's predictions. This interpretability is crucial for gaining the trust of clinicians and facilitating the integration of AI-based tools into clinical workflows.

**3.3.4. Innovative modeling approach.** Furthermore, our study is unique in its dual approach, comparing and combining sequential and static data modeling techniques. This comparison has revealed that while static models generally outperform sequential ones, each approach has its own strengths. By integrating these methods, we provide a more comprehensive analysis and understanding of the data.

## 3.4. Significance and contributions

Our study presents a novel approach to the prediction of BSI by addressing several critical challenges identified in previous works. One of the key innovations of our research is the effective handling of data imbalance, which we tackled by incorporating class weights and conducting a focused case study using a balanced dataset. Additionally, we derived meaningful and clinically relevant features from raw EHRs that are predictive of BSI, demonstrating the robustness of our feature engineering approach. Furthermore, we provided comprehensive comparisons between two types of data modeling techniques—sequential modeling using sequences of medical events and static modeling using a derived feature set. The use of SHAP values for interpretability further strengthens the clinical applicability of our models. By not depending on vital signs requiring real-time monitoring and data capture, our model can be applied more broadly across various healthcare settings. This includes environments where continuous monitoring might not be feasible, such as in lower-resource settings or outpatient care [51]. Using historical data minimizes disruptions to ongoing clinical workflows and does not necessitate immediate data collection [52]. Healthcare providers can utilize the predictive insights generated by our model without altering their routine patient assessments. Our framework's reliance on already available EHR data enhances its scalability. It can be deployed in numerous healthcare systems without additional infrastructure to capture and process vital signs, making advanced predictive tools more accessible to a broader range of healthcare

facilities. Utilizing comprehensive historical data allows for identifying infection risks before patients exhibit critical symptoms, potentially leading to preemptive treatments and better patient outcomes. This predictive capacity can transform care strategies from reactive to proactive, particularly in managing infections that can escalate rapidly if not addressed timely. The explainability aspect of the XAI approach increases trust among healthcare professionals by providing clear insights into the decision-making process of the AI model, which is crucial when clinical decisions are made [53].

### 3.5. Future works

In this study, we utilized SHAP values derived from the XGBoost model to explain the contributions of individual features to the model's predictions. The choice of XGBoost was driven by its balanced performance across multiple evaluation metrics, making it a robust model for generating interpretable feature importance scores. However, we acknowledge that SHAP values are inherently model-specific, meaning that the feature importance derived from XGBoost may differ from those derived from other models such as LightGBM, RF, or NN. While model-specific SHAP values provide valuable insights into how each model uses features to make predictions, there is an emerging interest in developing a more generalized feature importance measure that is independent of any particular model. One potential approach to achieve this would be to aggregate SHAP values across multiple models by computing the mean SHAP value for each feature. This aggregated SHAP value could offer a more holistic view of feature importance by capturing the consensus across different model architectures. However, this approach also presents certain challenges. Aggregating SHAP values across models could dilute the interpretability that SHAP is designed to provide, as different models may utilize features in fundamentally different ways based on their respective architectures and training processes. Therefore, while the concept of an aggregate SHAP value is appealing, it requires careful consideration to ensure that the resulting feature importance remains meaningful and interpretable. We believe that exploring the aggregation of SHAP values across multiple models could be a valuable direction for future research. Such an approach could enhance our understanding of feature importance by mitigating the biases associated with any single model and providing a more generalized perspective on how features contribute to predictions in complex ML tasks. This line of inquiry could lead to the development of more robust and interpretable ML models, particularly in critical applications such as clinical decision-making.

## 4. Materials and methods

### 4.1. Source of data

EHRs provide a longitudinal perspective of patients' interactions with hospital services. In Norway, which has predominantly public specialist healthcare, patients often have long and continuous histories within one hospital's records. This study harnessed EHRs from St. Olavs University Hospital, Trondheim, Norway, encompassing 35,591 patients with suspected BSI identified via physician-initiated BC between 2015 and 2020. The EHRs encompassed curated data from the inception of electronic records in 1999 until 2020, exclusively included hospital care episodes (excluding primary care and other specialist care episodes), ICU admission details, microbiology test results, laboratory test results, and patient demographics comprising of gender, date of birth, and date of death. Diagnoses and Procedures within these records were classified using the International Classification of Diseases, 10th Revision (ICD-10), facilitating standardized disease identification critical for the analytical models. This study adhered

to the 'transparent reporting of a multivariable prediction model for individual prognosis or diagnosis (TRIPOD) [54].

## 4.2. XBSI framework

The XBSI framework aggregated various data types from the raw EHRs, including demographics, laboratory tests, microbiology tests, discharge summaries, and ICU admissions, as depicted in Fig 5. This dataset underwent preprocessing, event log creation for sequential ML models, and feature engineering for static ML models before being transformed and scaled to facilitate the respective model development pipelines, following subsections details the various steps in the XBSI framework.

**4.2.1. Data preprocessing and transformation.** Initially stored in a Postgres database, the medical data was converted into CSV files to facilitate easier manipulation and access. Utilizing Python libraries such as Pandas and NumPy, the CSV files comprising raw EHRs were loaded as dataframes for processing. The discharge summaries required several data cleaning steps to ensure the quality and relevance of the data: Relevant patient information such as

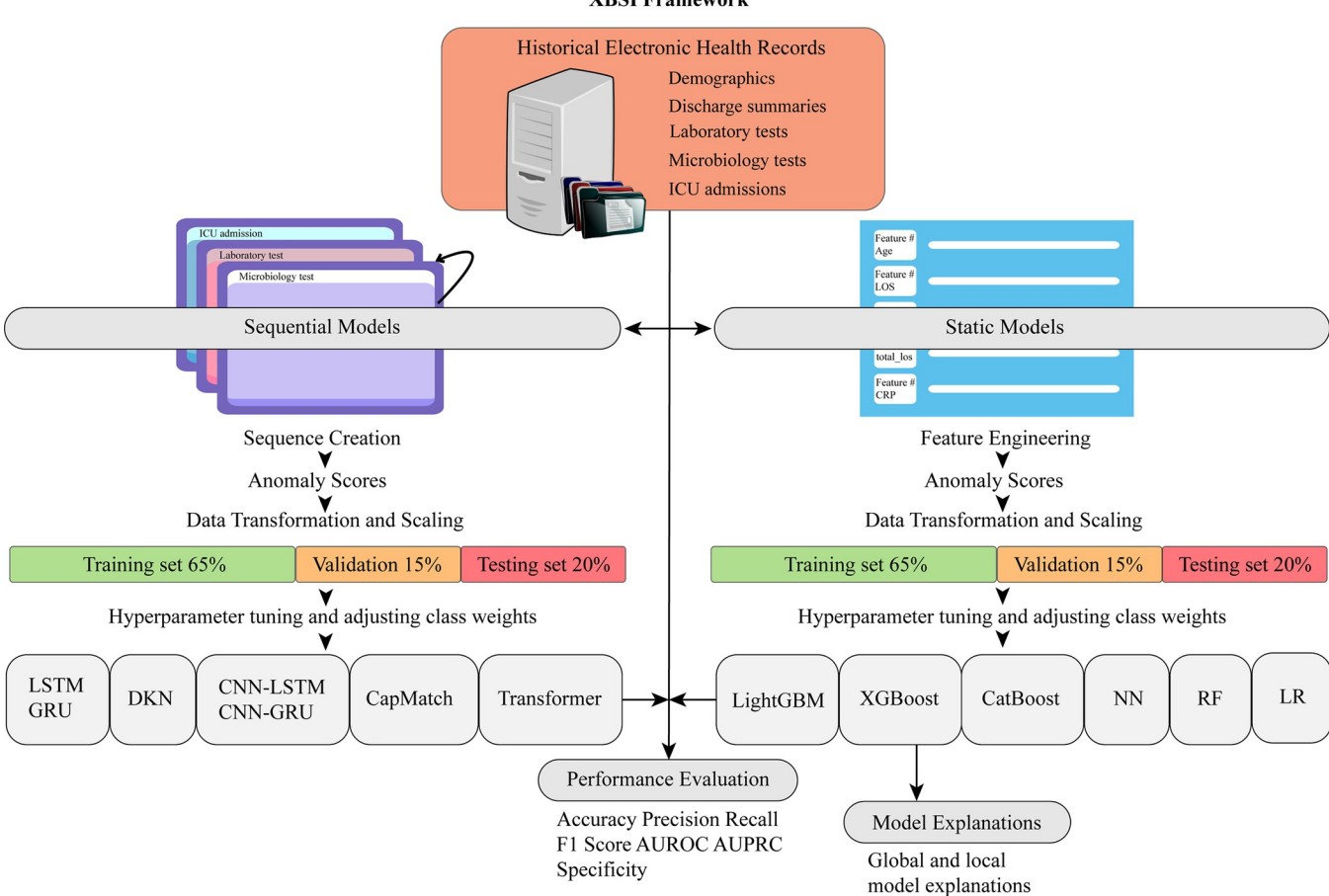

**Fig 5. Schematic overview of the XBSI framework.** The diagram illustrates the XBSI framework applied in the study, starting with the extraction of electronic health records (EHRs), which include demographics, laboratory tests, discharge summaries, microbiology tests, and ICU stays. The workflow bifurcates into two parallel processes: Sequence Creation and Feature Engineering, incorporating Anomaly Scores derived from the data. Subsequent steps include Data Transformation and Scaling, with the data split into training (at 65% and 80%, respectively), validation (at 15%), and testing (at 20%) subsets. The bottom layer of the workflow depicts the range of ML models, the performance metrics used to assess them, and the model explanations generated to study the feature importances.

identifiers, admission and discharge times, and diagnostic codes were retained. Instances of missing identifiers were addressed by replacing empty strings with empty values and removing these records. Data were organized by patient identifier and admission/discharge times to maintain coherent episode tracking. Non-standard characters within diagnostic codes, such as semicolons and commas, were standardized to spaces, and any duplicates were removed. The timestamps were converted into datetime format, facilitating the calculation of the LOS in hours for each episode. The duration of each ICU stay was calculated in hours, along with the total count of each type of hospital admission per patient. The request dates in laboratory and microbiology test results were standardized to datetime objects and used to create event logs of tests per patient. Specialized functions were designed to clean the laboratory and microbiology test table entries. This function performed tasks to remove any non-numeric characters, which could represent encoding errors or artifacts from data entry. It standardized decimal point characters by replacing commas with periods, necessary for consistent numerical representation across different regions that may use varying formats for decimal points. The microbiology test table was filtered to identify suspected BSI episodes, emphasizing BC tests. The results column was processed to standardize and clean the values, categorizing them as 'positive', 'negative', or 'contaminant' based on the results column. Four event logs were created for each patient ID, including discharge summaries, ICU admissions, laboratory tests, and microbiology tests. The following subsection describes the event logs.

**4.2.2. Event log description.** *Discharge Summaries Event Log*: Captures patient discharge information, including admission and discharge times, diagnostic and procedural codes, urgency, and care level codes.

*ICU Admissions Event Log*: Records details of each ICU stay, including the duration in hours, the total count of ICU admissions per patient, and the total length of ICU stays per patient.

*Laboratory Tests Event Log*: Includes results of various laboratory tests standardized and organized chronologically for each patient.

*Microbiology Tests Event Log*: Consists of microbiology test results, grouped by collection sample type, and categorizes them based on outcomes such as 'positive', 'negative', or 'contaminant'. Groups of microbiology tests categorized by collection sample type are given in the (List A in S2 File).

**4.2.3. Sequence creation.** The sequence creation process was implemented using the create_sequences function, designed to compile a comprehensive view of a patient's medical history over their entire recorded history. The create_sequences function systematically constructs a timeline of medical events for each patient and BC test. The medical events were merged from the four event logs. The filtered event logs are incorporated into a single dataframe, ensuring no information is lost. This step involves an outer join on patient ID and date, maintaining all records from each event log.

**4.2.4 Feature engineering.** This approach involved creating a dataset with attributes derived from diagnostic codes, procedure codes, and laboratory and microbiology test results. Laboratory test results were organized using pivot tables, ensuring a structured format for analysis. Tests such as 'bilirubin' (total, conjugated, and unconjugated), CRP, and 'lactate' (various measurements) were included, alongside WBC count (leukocytes), platelet count (thrombocytes), and blood gas measurements (pH, PO2). Similarly, microbiology test results were consolidated to reflect various sample types such as blood, urine, and other fluids, employing a dictionary mapping to streamline similar kinds. The resulting pivot table included columns for diverse samples, ranging from 'blood culture tests' to 'urine', 'feces', and 'nasal swabs'. A function was created to determine the aggregated result of BC tests for each patient group, considering the possibility of concurrent positive and contaminant results. This aggregation

provided a comprehensive view of the infection history per patient. Moreover, the history of prior positive, negative, and contaminant results was calculated and added to the dataset, offering a valuable perspective on the patient's previous encounters with infections. Comorbidities were extracted and processed to identify unique diseases from patient records. New columns were created for each disease, and the counts were updated based on patient history. Finally, aggregate columns were added for diagnostic and procedure codes to calculate cumulative sums. This method enabled capturing the cumulative history of medical conditions and procedures for each patient. All the empty values representing the absence of a condition or measurement were filled with zero, and "0" is not interpreted as a value.

**4.2.5. Model development.** This section outlines the development and evaluation of ML models. The process bifurcates into the sequence creation and feature engineering pipelines for the sequential and static ML pipelines respectively. An Isolation Forest model was implemented to detect anomalies within the data. This model was trained exclusively on normal data (negative BC episodes) and then used to compute anomaly scores for the training, validation, and test sets. These scores were normalized and appended to the original dataset to serve as additional features, enhancing the model's ability to distinguish between normal and abnormal patterns. Given the sequential and tabular nature of the input data for the respective ML pipelines, the necessary transformation steps were employed to prepare it for the corresponding learning algorithms. The data was reshaped and scaled using the functions from the sklearn library. This normalization step is crucial for models sensitive to input features' scale. After scaling, the data was reshaped to its original form, ensuring compatibility with the models. Sequential ML models included Long Short-Term Memory (LSTM), Gated Recurrent Unit (GRU), Convolutional Neural Network-Long Short-Term Memory (CNN-LSTM), Convolutional Neural Network-Gated Recurrent Unit (CNN-GRU), Transformer model, and two additional State-Of-The-Art (SOTA) ML models, Densely Knowledge-aware Network (DKN) and CapMatch (**S**emi-Supervised Contrastive Transformer Capsule with Feature-Based Knowledge Distillation for Human Activity Recognition). These models were trained on sequences of medical events to capture temporal dependencies within the data. On the other hand, the static ML models, which treat data points as independent and identically distributed, comprised of Light Gradient Boosting Machine (LightGBM), eXtreme Gradient Boosting (XGBoost), Categorical Boosting (CatBoost), Artificial Neural Network (ANN), Random Forest (RF), and Logistic Regression (LR). Detailed mathematical formulation and computational complexity for each ML model are given in (Section 1.1 and 1.2 in S1 File).

**4.2.6. Model validation.** For cross-validation, we chose a 5-fold cross-validation scheme. This technique involves dividing the entire dataset into 'k' equally sized subsets or folds. The model is then trained on 'k-1' folds and tested on the remaining fold. This process is repeated 'k' times with a different fold as the validation set. In addition to cross-validation, we also validated our model on an independent test set, which was separated from the dataset at the outset and not used during the training phase. The test set comprised the most recent 20% of the BC episodes, while the earliest 80% of BC episodes were assigned as training cohorts. Further, the data was divided into training and validation sets, with 15% of the data allocated for validation. This additional split allowed for tuning hyperparameters and assessing the model's performance during training. Each model was evaluated using a variety of metrics to evaluate model performance, including accuracy, precision, recall, F1-score, specificity, the Area Under Precision-Recall (PR) Curve (AUPRC), and the Area Under the Receiver Operating Characteristic (AUROC). Detailed information and mathematical formulations of each performance metric are given in (Section 1.3 in S1 File).

**4.2.7. Model explanations.** We employ global and local explanation methods to interpret our ML models' outputs to provide comprehensive insights into feature importances. For

global explanations, we generate SHAP (SHapley Additive exPlanations) summary plots [55]. These plots aggregate SHAP values to illustrate the average influence of each feature on the model output, ranked by significance. This method, rooted in game theory, decomposes a prediction into the contribution of each feature, providing a transparent view of the predictive process. SHAP's suitability for tree-based models was enhanced by its efficient computation of exact SHAP values using the TreeExplainer algorithm [56], significantly reducing computational complexity by exploiting the structural properties of decision trees. For local explanations, we utilize waterfall plots and force plots to detail the contribution of each feature to specific predictive outcomes. Waterfall plots provide a step-by-step breakdown of how each feature's value contributes to the final prediction, starting from the base value (the average model output across all data points) and adding the effect of each feature sequentially. This visualization helps understand the decision-making process for individual predictions, which is crucial for clinical validation and personalized patient insights. Force plots, another local interpretability tool, display how each feature's value pushes the model's prediction higher or lower, particularly useful for individual patient assessments. These plots highlight each feature's positive or negative contribution towards the final prediction, allowing healthcare providers to grasp the underlying reasons for a model's decision on a case-by-case basis.

### 4.3. Participants and outcome

All adults (aged $\geq$ 18 years) who had at least one BC episode during their hospital stay or visit, which was identified and ordered by a physician on the grounds of suspicion of a BSI. The primary outcome was whether a BC episode was positive or negative for bacteremia. A BC episode was defined as a distinct nonoverlapping 24-hour period in which one or more BC tests were ordered [43]. If one or more results within a BC episode were positive, then the BC episode was considered positive. BC results with contaminants were considered negative results [57]. The list of microbes considered as contaminants is given in (Table F in S2 File).

### 4.4. Predictors

We used data available till the date of a BC episode for training the ML prediction models. The predictors included age, sex, results of the recent laboratory test values, previous positive microbiology tests, count of co-morbidities, diagnostic and procedural codes, and total ICU stays. The most common laboratory tests were bilirubin, CRP, creatinine, leukocytes, and thrombocytes. The counts of prior positive results of microbiology tests grouped by their collected sample type were calculated and used as predictors of previous history of infections (S1 File). From the medical history, the predictors included counts of the occurrences of different ICD-10 diagnostic and the Nordic Medico-Statistical Committee (NOMESCO) Classification of Surgical Procedures (NCSP), codes and Classification of Medical Procedures (NCMP) codes, were classified according to the initial character (alphabetic), corresponding to the various chapters of the ICD-10. Each patient record was expanded with new columns for the counts of the ICD codes in the recent and complete history corresponding to each character, incrementing the count for each instance where a character led the code. For calculation of diagnostic and procedural code counts in the recent episode, the current episode and any admissions or visits within one month of the BC test date were merged into the current medical episode. The LOS feature stored the length of stay of the most recent hospital episode until BC. The total LOS feature stored the value of the cumulative hospital length of stay per patient. The description of each predictor and its mean across the dataset is given in Table D in S2 File. Further stratification was conducted by categorizing diagnostic codes into disease groups pertinent to clinical significance, such as 'explicit sepsis', 'infection', and 'organ dysfunction'. The

table depicting the diagnostic codes selected for different disease groups is given in Table E in S2 File.

## 4.5. Prediction task modeling

We undertook prediction modeling using two distinct datasets: a static model dataset, $X$, and a sequential model dataset, $Y$. The static dataset, $X$, encompasses labeled aggregated, patient-specific information available till the day of the BC episode. Using $X$, we trained static models $M_{sta}$ to predict the likelihood of a positive BC episode. The sequential dataset, $Y$, contains labeled sequences of medical events per patient compiled from the event logs for each BC test. The sequential model $M_{seq}$ is developed using $Y$ to predict the likelihood of a positive BC episode.

## 4.6. Case study: Patient selection process and XBSI workflow

**4.6.1. Patient selection criteria.** The selection criteria included the following steps:

*Data Segregation*: The original dataset was divided into two groups: positive cases (those with BSI) and negative cases (those without BSI).

*Deduplication*: To ensure each patient was only represented once, duplicate entries were removed based on patient identifiers.

*Downsampling*: From each group, 500 patients were randomly selected to create a balanced dataset. This process was carefully managed to avoid any potential bias, ensuring that both groups were representative of the overall population.

*Data Shuffling*: The final dataset was shuffled to eliminate any ordering effects, resulting in a balanced and randomized dataset comprising 1000 patients, equally split between positive and negative BSI cases.

**4.6.2. XBSI workflow.** The XBSI framework was employed to generate features from the historical medical data of the selected patients and subsequently train both static and sequential ML models. The workflow followed these steps:

*Feature extraction/engineering*: Relevant features/predictors were extracted from the patient's medical history, including demographic data, clinical measurements, and prior diagnoses. For static models, this involved aggregating features that represented the patients' state at the time of the BC test.

*Sequence generation*: For the sequential models, sequences of medical events leading up to the BC test were generated. These sequences included time-ordered events such as hospital and ICU admissions, and laboratory, and microbiology test results, providing a temporal context to the model.

*Model development*: Static models and sequential models were trained using the generated features and sequences, respectively. The models were evaluated using standard metrics. The balanced dataset helped in mitigating the effects of class imbalance, leading to more robust and reliable model performance.

*Interpretation of results*: SHAP values were used to interpret the models' predictions.

## 4.7. Computational complexity analysis

To comprehensively evaluate the ML models, we conducted an analysis of their computational complexity. This analysis is essential to understand the trade-offs between model performance and the resources required for model training and inference, particularly in a clinical setting where computational efficiency can directly impact real-time decision-making.

**4.7.1. Sequential models.** *LSTM and GRU Models*: The computational complexity of LSTM and GRU models primarily depends on the number of parameters and the sequence

length. The time complexity for both models is $O(n{\times}d{\times}h)$, where $n$ is the number of time steps (sequence length), $d$ is the dimensionality of the input, and $h$ is the number of hidden units. GRU models are slightly more efficient than LSTM models due to their simpler architecture, as GRUs have fewer gates, reducing the number of operations required per time step.

*CNN-LSTM and CNN-GRU Models*: These hybrid models combine convolutional operations with LSTM or GRU layers. The convolutional layers' complexity is $O(f{\times}k{\times}d)$, where $f$ is the number of filters, $k$ is the kernel size, and $d$ is the input dimension. This is followed by the sequential layer complexities mentioned above. Although these models capture both spatial and temporal features, they are computationally more intensive due to the convolutional operations.

*Transformer Model*: The Transformer model's complexity is $O(n^2{\times}d)$ for the multi-head self-attention mechanism, where $n$ is the sequence length and $d$ is the dimension of the model. This quadratic dependence on the sequence length makes the Transformer model more computationally expensive compared to RNN-based models like LSTM and GRU, especially for longer sequences. However, the model's ability to handle parallelization can offset this during training, leading to faster convergence times.

*DKN Model*: The DKN model integrates both sequential and knowledge graph information, which increases its complexity. The complexity for this model can be represented as ($n{\times}d{\times}h$ $+g{\times}e{\times}h$), where $n$ is the sequence length, $d$ is the input dimension, $h$ is the number of hidden units, $g$ is the number of graph nodes, and $e$ is the embedding dimension of the knowledge graph. The additional complexity arises from integrating the knowledge graph, which requires additional computation to align the graph embeddings with the sequence data.

*CapMatch Model*: The CapMatch model uses a unique method for handling missing and uncertain data by employing capsule networks and matching networks. The complexity of this model is determined by both the capsule networks $O(c{\times}d{\times}r)$, where $c$ is the number of capsules, $d$ is the dimension of the capsule output, and $r$ is the routing iterations, and the matching network $O(m{\times}d{\times}s)$, where $m$ is the memory size, $d$ is the input dimension, and $s$ is the support set size. This dual architecture, while powerful for handling complex patterns and uncertainty, is computationally demanding, particularly in terms of memory usage and inference time.

**4.7.2. Static models.**    *Tree-based Models*: The computational complexity of the RF model is generally $O(t{\times}d{\times}\log n)$, where $t$ is the number of trees, $d$ is the depth of each tree, and $n$ is the number of samples. This complexity reflects the time required to construct the ensemble of decision trees and make predictions. RF is known for its robustness and ability to handle large datasets effectively, but the model's complexity increases with the number of trees and the depth of each tree. The other tree-based models (XGBoost, LightGBM, CatBoost) have computational complexity f similar to RF but with additional optimizations. XGBoost typically demands more computational resources due to its extensive regularization techniques, while LightGBM is optimized for speed and memory usage through histogram-based techniques and leaf-wise growth strategies. CatBoost, on the other hand, efficiently handles categorical data without requiring extensive preprocessing, reducing overall computation time.

*Artificial Neural Network (ANN)*: The complexity of the ANN used in this study is $O(l{\times}n{\times}d)$, where $l$ is the number of layers, $n$ is the number of neurons per layer, and $d$ is the input dimension. The model's fully connected nature increases the number of parameters, leading to higher computational demands, especially as the network depth increases.

*Logistic Regression (LR)*: Logistic regression has a linear computational complexity of $O(n{\times}d)$, where $n$ is the number of samples and $d$ is the number of features.

*Summary of Computational Trade-offs*: The computational complexity of the models presents a clear trade-off between performance and resource efficiency. Sequential models like LSTM, GRU, DKN, and Transformers offer robust performance for time-series data, but their

higher computational costs necessitate careful consideration in deployment, especially in resource-constrained environments. The CapMatch model, while highly effective at handling uncertainty and missing data, also introduces significant computational demands due to its complex architecture. The RF model, although computationally expensive during training, is often faster during inference, making it well-suited for real-time applications. Other static models, particularly the tree-based ensemble models, provide a balanced approach, offering high accuracy with moderate computational demands. Logistic regression remains the most efficient in terms of computational resources, though at the expense of predictive performance.

### 4.8. Statistical analysis

The mean values for the significant features across the two classes were computed, providing an understanding of how each feature varies with the BC episode results. Statistical analysis was performed using independent t-tests to compare the means of each feature between two independent groups labeled by blood culture test results [58]. A Pearson correlation matrix was constructed for the significant features to examine the strength and directionality of the relationships between them [59]. The resulting coefficients were visualized using a heatmap. The results of the statistical analysis are given in the (S1 File).

### 4.9. Ethics statement

This study utilizes a de-identified dataset comprising EHRs from the St. Olavs University Hospital, Trondheim, Norway. The data was accessed and analyzed through a secure private cloud platform. The use of the EHRs in this project has been approved by the Regional Committee for Medical and Health Research Ethics (REK) in Central Norway by REK no. 2020/26184.

## 5. Conclusions

In this study, we successfully developed and validated a robust predictive framework for BSI using historical EHRs. Our approach, which integrated both sequential and static ML models, demonstrated that static models outperformed sequential models in terms of predictive performance. This finding underscores the importance of data enhancement and tree-based ML models, especially when dealing with complex clinical datasets [60]. We also addressed key challenges in BSI prediction, such as data imbalance and model interpretability. By applying class weighting and utilizing SHAP values, we enhanced the model's ability to identify early predictors of risk of BSI. The presented XBSI framework is the first to significantly enhance predictive analysis by integrating information stored as diagnostic and procedural ICD-10 codes. This novel approach diverges considerably from traditional real-time monitoring systems, emphasizing incorporating comprehensive historical patient data. The key predictors include the count of ICD-10 codes for infectious diseases, kidney and urinary disorders, bilirubin, creatinine, leukocytes, thrombocytes, age, prior positive BC tests, and the total length of hospital stay until the BC test. To further enhance the generalizability of our findings and adoption of our framework, we need to validate our framework on administrative datasets of hospitals outside Norway and include more diverse data sources, such as, genotypes and phenotypes, in future works.

## Supporting information

**S1 File. Supplementary methods.** Additional materials and methods, including mathematical formulations for the ML models and performance metrics.
(DOCX)

**S2 File. Supplementary results.** Contains additional results on statistical analysis and description of predictors. **Fig A:** Correlation matrix of clinical predictors **Table A.** Comparison of the predictors (Top 25 most influential) between the two classes. **Table B.** Statistical significance of clinical Predictors. **Table C.** Correlation coefficients of most correlated Features. **Table D.** List of all the predictors, their description and their average values across the dataset. **Table E.** List of ICD-10 codes used to classify selected diseases. **Table F.** List of microbes identified as contaminants **List A.** Groups of various microbiology tests.
(DOCX)

## Acknowledgments

We want to thank the Mid-Norway Centre for Sepsis Research researchers for their valuable discussions and feedback.

## Author Contributions

**Conceptualization:** Rajeev Bopche, Jan Kristian Damås, Øystein Nytrø.

**Data curation:** Rajeev Bopche, Lise Tuset Gustad.

**Formal analysis:** Rajeev Bopche, Birgitta Ehrnström, Jan Kristian Damås, Øystein Nytrø.

**Funding acquisition:** Lise Tuset Gustad, Jan Kristian Damås, Øystein Nytrø.

**Investigation:** Rajeev Bopche, Jan Egil Afset, Birgitta Ehrnström, Jan Kristian Damås.

**Methodology:** Rajeev Bopche.

**Project administration:** Rajeev Bopche, Jan Kristian Damås, Øystein Nytrø.

**Resources:** Rajeev Bopche.

**Software:** Rajeev Bopche.

**Supervision:** Jan Egil Afset, Birgitta Ehrnström, Jan Kristian Damås, Øystein Nytrø.

**Validation:** Rajeev Bopche.

**Visualization:** Rajeev Bopche.

**Writing – original draft:** Rajeev Bopche.

**Writing – review & editing:** Rajeev Bopche, Birgitta Ehrnström.

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
