## [Decision Letter · Decision Letter 0]

14 Jun 2024

PDIG-D-24-00143

Advancing Bloodstream Infection Prediction Using Historical Electronic Health Records

PLOS Digital Health

Dear Dr. Bopche,

Thank you for submitting your manuscript to PLOS Digital Health. After careful consideration, we feel that it has merit but does not fully meet PLOS Digital Health's publication criteria as it currently stands. Therefore, we invite you to submit a revised version of the manuscript that addresses the points raised during the review process.

Please submit your revised manuscript within 60 days Aug 13 2024 11:59PM. If you will need more time than this to complete your revisions, please reply to this message or contact the journal office at digitalhealth@plos.org. Please include the following items when submitting your revised manuscript:

We look forward to receiving your revised manuscript.

Kind regards,

Qihuang Zhang

Academic Editor

PLOS Digital Health

Journal Requirements:

1. Please provide separate figure files in .tif or .eps format only and remove any figures embedded in your manuscript file. Please also ensure that all files are under our size limit of 10MB.

2. We have noticed that you have uploaded Supporting Information files, but you have not included a list of legends. Please add a full list of legends for your Supporting Information files after the references list.

3. In the online submission form, you indicated that "The jupyter notebooks implementing the data preprocessing, sequence creation, feature engineering and model development pipeline are available through the corresponding author. The processed and transformed final datasets, and the list of derived features are available through the corresponding author on a reasonable request". 

3. Uploaded as supplementary information.

Additional Editor Comments (if provided):

I share a similar question as the reviewer regarding the model's performance. Also, the speaker should examine the interpretation of the SHAP values. For example, a lower SHAP value represents a negative influence on the prediction rather than indicating that the predictor is not important (e.g., 'anomaly_score').

Reviewers' comments:

Reviewer's Responses to Questions

**Comments to the Author**

1. Does this manuscript meet PLOS Digital Health’s publication criteria? Is the manuscript technically sound, and do the data support the conclusions? The manuscript must describe methodologically and ethically rigorous research with conclusions that are appropriately drawn based on the data presented.

Reviewer #1: Yes

Reviewer #2: Yes

2. Has the statistical analysis been performed appropriately and rigorously?

Reviewer #1: Yes

Reviewer #2: Yes

3. Have the authors made all data underlying the findings in their manuscript fully available (please refer to the Data Availability Statement at the start of the manuscript PDF file)?

Reviewer #1: Yes

Reviewer #2: Yes

4. Is the manuscript presented in an intelligible fashion and written in standard English?

Reviewer #1: Yes

Reviewer #2: Yes

5. Review Comments to the Author

Reviewer #1: The research work reported is interesting. There are some major issues that the authors consider addressing before acceptance: 

1. [Case Study] To further understand the proposed method, the authors may provide a case study. 

2. [Motivation and Contribution] As known, there are some existing relevant methods in the community. The authors should include these studies in the manuscript. What are the improvements of the proposed method compared with these existing methods? What problems did the previous works exist? How to solve these problems? The authors may consider analyzing the problems of the previous works and how to address these problems in the manuscript. Please explain that.

3. [SOTA] The authors should compare the proposed method with some recent algorithms.

4. [Mathematics] The authors consider some mathematical reasonings in the manuscript.

5. [Research Gap] The research gaps should be highlighted and explained in the manuscript.

6. [State Similar Studies] There are some existing algorithms that have some merits similar to the proposed method. The authors should explain the differences between the proposed method and existing algorithms, including Densely Knowledge-aware Network for Multivariate Time Series Classification, CapMatch: Semi-supervised Contrastive Transformer Capsule with Feature-based Knowledge Distillation for Human Activity Recognition, DTCM: Deep Transformer Capsule Mutual Distillation for Multivariate Time Series Classification, and Deep Contrastive Representation Learning With Self-Distillation.

7. [Computational Complexity] Computational complexity should be analyzed and compared in the manuscript.

Reviewer #2: 1. In line 246, does it indicate that the “Temporal ML models” utilizing additional time variable compared to “static models”? If so, is there explanation about why static model works better than the sequential model incorporating additional information.

2. In the model performance section, a bunch of the model evaluations were used, is there a preferred metric here for bloodstream infection prediction? Especially given the fact of the imbalance feature of the dataset? Have you tried to apply your models on the balanced dataset and see if the model performance/SHAP changed?

3. In Table 2, I am surprised that the precision of RF is 0.8510, which surpass other models including other tree based models, but a low recall of 0.2201; Similar situation applies for NN but not as extreme as RF. Just wonder if it is still affected by the unbalance nature of the dataset, might be good to run a sensitivity analysis through adjusting the weights of the minority group (or resampling), which could potentially make precision/recall score more stable. Maybe Ensemble technique could be used to provide an overall performance.

4. Since Supplementary Table 3 shows some features are highly correlated, it would be great to explain whether/to what extend it affect the shap value, so that the independence assumption are still met.

5. The current SHAP value is from XGBoost model, is it because it has the “balanced performance” (indicated by line 370)? Just wonder if it is possible to have overall “feature importance” ignoring which particular model is picked.

6. Overall for the XBSI framework, in Figure 1, due to the performance for difference model type dynamic/static, does it suggest static models are suitable with the prediction task for the future.

6. PLOS authors have the option to publish the peer review history of their article (what does this mean?). If published, this will include your full peer review and any attached files.

**Do you want your identity to be public for this peer review?** For information about this choice, including consent withdrawal, please see our Privacy Policy.

Reviewer #1: No

Reviewer #2: No

---

## [Decision Letter · Decision Letter 1]

29 Sep 2024

Leveraging Explainable Artificial Intelligence for Enhanced Prediction of Bloodstream Infections Using Historical Electronic Health Records

PDIG-D-24-00143R1

Dear Mr. Bopche,

We are pleased to inform you that your manuscript 'Leveraging Explainable Artificial Intelligence for Enhanced Prediction of Bloodstream Infections Using Historical Electronic Health Records' has been provisionally accepted for publication in PLOS Digital Health.

Best regards,

Qihuang Zhang

Academic Editor

PLOS Digital Health

Reviewer Comments (if any, and for reference):

Reviewer's Responses to Questions

**Comments to the Author**

The authors added more clarification of how the temporal part contribute to the static models, clarified the the advantage of the static models over the sequential models providing more sensitivity analysis and provided a comprehensive analysis contributing the bloodstream infection prediction.

The portion that may need minor corrections:

1. In section 3.4 line 611, it seems like the sentence “Our findings indicate that while static models generally outperformed sequential models.” Is not complete otherwise might be good to delete “while”.

2. The horizontal Prediction 2 part in Figure 4 has label and value overlapped, it would be better to modify the layout.

 **********